**METHOD**

# VarID2 quantifies gene expression noise dynamics and unveils functional heterogeneity of ageing hematopoietic stem cells

Reyna Edith Rosales-Alvarez[1,2,3] , Jasmin Rettkowski[4,5], Josip Stefan Herman[1], Gabrijela Dumbović[1], Nina Cabezas-Wallscheid[4,6] and Dominic Grün[1,7*]

*Correspondence:
dominic.gruen@uni-wuerzburg.de

[1] Würzburg Institute of Systems Immunology, Max Planck Research Group at the Julius-Maximilians-Universität Würzburg, Würzburg, Germany
[2] International Max Planck Research School for Immunobiology, Epigenetics, and Metabolism (IMPRS-IEM), Freiburg, Germany
[3] Faculty of Biology, University of Freiburg, Freiburg, Germany
[4] Max Planck Institute of Immunobiology and Epigenetics, Freiburg, Germany
[5] Spemann Graduate School of Biology and Medicine (SGBM), Freiburg, Germany
[6] CIBSS-Centre for Integrative Biological Signaling Studies, University of Freiburg, Freiburg, Germany
[7] Helmholtz Institute for RNA-Based Infection Research (HIRI), Helmholtz-Center for Infection Research (HZI), Würzburg, Germany

## Abstract

Variability of gene expression due to stochasticity of transcription or variation of extrinsic signals, termed biological noise, is a potential driving force of cellular differentiation. Utilizing single-cell RNA-sequencing, we develop VarID2 for the quantification of biological noise at single-cell resolution. VarID2 reveals enhanced nuclear versus cytoplasmic noise, and distinct regulatory modes stratified by correlation between noise, expression, and chromatin accessibility. Noise levels are minimal in murine hematopoietic stem cells (HSCs) and increase during differentiation and ageing. Differential noise identifies myeloid-biased Dlk1+ long-term HSCs in aged mice with enhanced quiescence and self-renewal capacity. VarID2 reveals noise dynamics invisible to conventional single-cell transcriptome analysis.

**Keywords:** Gene expression noise, Single-cell RNA sequencing, Stem cell differentiation, Cell sate variability, Ageing, Hematopoietic stem cells, Machine learning, Mathematical modeling

## Background

Single-cell genomics has become a powerful method of choice for the identification of cell types and for the inference of tissue composition at single-cell resolution [1–3]. The reconstruction of cell state manifolds paired with pseudotime analysis facilitates the derivation of ancestral relations between cell states and enables the prediction of cellular differentiation trajectories [4]. However, such inference methods heavily rely on transcriptome similarity and are therefore limited in their ability to capture control mechanisms of cell fate choice driven by lowly expressed genes. For example, single-cell lineage tracing by random cellular barcoding revealed early lineage priming of hematopoietic stem cells (HSCs) [5], which remained undetected when relying on single-cell RNA-sequencing (scRNA-seq) data alone. A known problem for the quantification of subtle

expression changes, in particular for lowly expressed genes, is the substantial level of technical variability masking genuine biological variability, or noise [6, 7].

Gene expression noise is prevalent in unicellular organisms [8, 9] and can underlie bistable systems such as the *E. coli* lac operon [10]. Increased variability of gene expression has been observed during in vitro differentiation of embryonic stem cells [11] or upon reprogramming of induced pluripotent stem cells [12], yet its role during cell fate decision within multilineage systems in vivo is underexplored [13, 14].

Although scRNA-seq allows noise quantification within homogenous cell populations [6, 15–18], available methods cannot capture biological noise dynamics at high resolution across complex cell state manifolds.

We recently proposed VarID as a method for quantifying local gene expression variability in cell state space, which eliminates the mean dependence of gene expression variability but does not explicitly distinguish technical and biological sources of noise [19]. We here introduce VarID2 to overcome this major limitation by modeling defined sources of technical noise in local cell state neighborhoods, facilitating the inference of actual biological variability. We demonstrate that VarID2 predicts biological noise levels consistent with state-of-the-art Bayesian noise models [18, 20], which are only applicable to pairwise comparisons of large homogenous cell populations profiled by scRNA-seq, and, hence, do not permit the investigation of noise dynamics during cellular differentiation.

VarID2 analysis of human peripheral blood mononuclear cells (PBMCs) indicates a general increase of transcriptome variability in the nucleus compared to the cytoplasm, and the relation between chromatin accessibility, gene expression, and noise uncovers distinct modes of gene regulation.

Noise quantification within the murine hematopoietic system reveals minimal noise in HSCs which increases upon differentiation. The hematopoietic system is known to be affected by ageing, with a gradual functional decline of HSCs and an emerging myeloid bias in the bone marrow [21]. It is still unclear to what extent the age-dependent decline of the hematopoietic system can be attributed to an emerging HSC heterogeneity, resulting from age-related changes of cell-intrinsic properties, or from a changing bone marrow microenvironment. By applying VarID2, we observed increased transcriptome variability in HSCs of aged mice. The top noisy gene in aged HSCs, *Dlk1*, facilitates the discrimination of two sub-populations of HSCs which are almost indistinguishable on the global transcriptome level, yet exhibit clear differences in terms of quiescence, self-renewal capacity, and myeloid bias. We argue that age-related emergence of Dlk1+HSCs with cell-intrinsic myeloid bias could contribute to the age-dependent change of bone marrow composition. Hence, we demonstrate that single-cell resolution analysis of gene expression noise can yield fundamentally new biological insights. VarID2 was integrated into our RaceID toolkit for single-cell analysis publicly available on CRAN.

## Results

### Modeling local gene expression variability in cell state space

To model local gene expression variability in cell state space, we are building upon our previous VarID method [19]. VarID constructs a pruned *k*-nearest neighbor (knn) graph in cell state space and tests transcript count differences between the

"central" cell and each of its neighbors against a background model. In VarID2, this background model is defined as a negative binomial with a local mean of raw unique molecular identifier (UMI) counts and a corresponding standard deviation obtained from a local fit of the mean–variance dependence across all genes (see "Methods"). To overcome the lack of VarID in resolving technical and biological noise components and to estimate genuine biological variability, we reasoned that two sources of noise dominate the observed UMI count variance measured for each gene in such local neighborhoods. At low expression, sampling noise, i.e., binomial variance captures the average trend (Fig. 1a): in this regime, the dependence of the coefficient of variation (CV) on the mean follows a line of slope $-1/2$ in logarithmic space. At high expression, the CV-mean dependence saturates and approaches a baseline variance level. As described previously [6], this baseline is determined by the shared variability affecting all genes equally, which we refer to as total UMI count variability. Major sources of this noise component are cell-to-cell differences in sequencing efficiency and cell volume [6]. We inferred this noise component by fitting a Gamma distribution to the total UMI count distribution across cells in each local neighborhood. The resulting Poisson-Gamma mixture corresponds to a negative binomial distribution, describing the UMI counts $X_{i,j}$ for each gene $i$ across a neighborhood with central cell $j$. The parameters of this distribution are the local mean $\mu_{i,j}$ of the raw UMI counts, and dispersion parameter $r^t_{\ j}$ given by the rate parameter of the Gamma distribution ("Methods").

The remaining residual variability in excess of these two major sources can be summarized into an additional dispersion parameter $\varepsilon_{i,j}$ (Fig. 1b). We refer to this residual variability as biological noise since it captures gene-specific deviations from the global trend determined by sampling variability and total UMI count variance.

By applying VarID2 to scRNA-seq data of mouse Kit+ hematopoietic progenitors [22] comprising major branches of erythrocyte and neutrophil progenitors, we observed that $r^t_{\ j}$ indeed varies substantially between distant neighborhoods in cell state space. Thus, a local noise model is required to quantify this noise component for heterogeneous cell populations (Fig. 1c, d). Since a maximum likelihood (ML) fit of the biological noise $\varepsilon_{i,j}$ led to inflated noise estimates for lowly expressed genes, we incorporated a weakly informative Cauchy prior and performed maximum a posterior (MAP) estimation of $\varepsilon_{i,j}$, which eliminated the inflation ("Methods"; Fig. 1e and Additional file 1: Fig. S1a). To test our noise model quantitatively, we simulated cell neighborhoods with defined technical and biological noise levels based on gene expression parameters from [22] ("Methods" and Additional file 1: Fig. S1b). We optimized the scale parameter $\gamma$ of the Cauchy prior by jointly matching the median and minimizing the standard deviation of the estimates compared to the simulated ground truth (Additional file 1: Fig. S1c). This analysis demonstrates the accuracy of our noise estimates across three different noise levels, as well as the absence of a systematic mean–variance dependence (Fig. 1f). Notwithstanding, noise estimates for lowly expressed genes tend to deviate from the ground truth due to limited statistical power. In order to filter out lowly expressed genes, we tested different thresholds of gene expression, and we assessed if $\varepsilon$ estimates were within a twofold confidence interval around the ground truth. We suggest an optional expression threshold between 0.3 and 0.4, since it preserves around 60% of genes within the dataset and 60

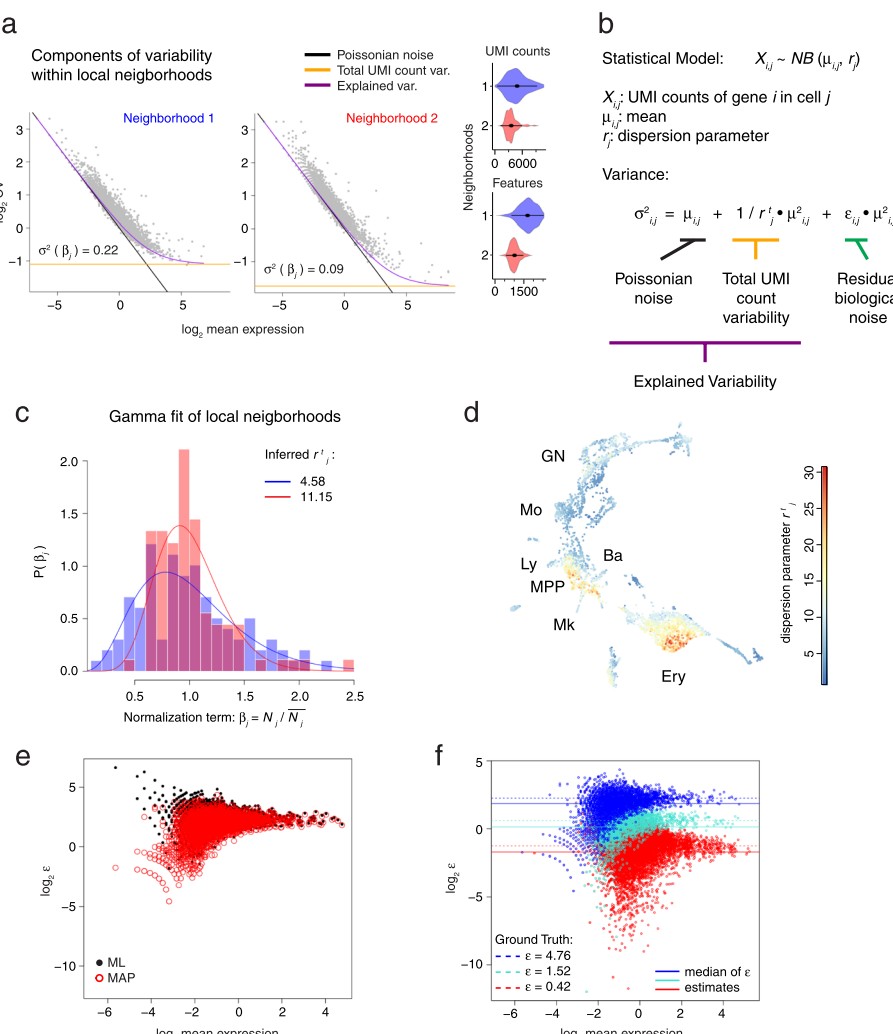

**Fig. 1** Local decomposition of gene expression noise in cell state space. **a** Coefficient of variation as a function of the mean expression on logarithmic scale. The explained variability and its components, Poissonian noise and total UMI count variability, are highlighted. Dot plots correspond to two individual neighborhoods of 101 cells each from a Kit+ hematopoietic progenitor dataset [22]. Violin plots to the right show the distribution of UMI counts and the number of detected features per cell barcode for each of the individual neighborhoods. **b** Negative binomial model for the UMI counts $X_{i,j}$. The variance is split into three components: Poissonian noise, total UMI count variability, and residual biological noise. **c** Estimation of the dispersion parameter $r_j^t$ for the two individual neighborhoods shown in **a**. Mean-normalized total UMI counts $\beta_j$ are fitted by a Gamma distribution, with shape parameter $\alpha_j^t$ equal to the dispersion parameter $r_j^t$ in **b**. **d** UMAP plot highlighting $r_j^t$ estimates across the hematopoietic progenitor dataset. MPP, multipotent progenitors; Ly, lymphocytic; Mo, monocytic; GN, granulocytic neutrophil; Ba, basophilic; Mk, megakaryocytic; Ery, erythroid. **e** Comparison of $\varepsilon$ estimates obtained by maximum likelihood (ML) estimation (black) and maximum a posteriori (MAP) estimation (red). A simulated dataset with three levels of gene expression noise was used (see "Methods" and Additional file 1: Figure S1b). Here, only $\varepsilon$ estimates corresponding to the highest noise level are shown. **f** $\varepsilon$ estimates for the simulated dataset with three different biological noise levels ("Methods"). Colors highlight groups of genes with different simulated biological noise levels (low, medium, or high). Simulated ground truths of noise values (dashed lines), and median values of the $\varepsilon$ estimates (solid lines) are indicated for each group. Hyperparameter $\gamma = 1$

to 90% of those have noise estimates within the ground truth confidence interval (Additional file 1: Fig. S1d).

In order to make VarID2 scalable, we restricted the model to MAP estimation of the residual noise parameter $\varepsilon$. BASiCS has been introduced as a full Bayesian noise model with multiple parameters [18, 20], yet this model is computationally expensive and application to a larger number of local neighborhoods is infeasible. Moreover, the biological noise parameter of BASiCS [20] is defined as the residual over-dispersion from the average mean–variance dependence. In contrast, VarID2 assigns a clear interpretation to $\varepsilon_{i,j}$ as a residual after deconvoluting defined noise components. Reassuringly, $\varepsilon_{i,j}$ is highly correlated with BASiCS' over-dispersion parameter (Pearson's correlation coefficient 0.85) with diminished correlation of the ML estimate (Pearson's correlation coefficient 0.79), supporting our choice of the prior (Additional file 1: Fig. S1e, f). Hence, VarID2 overcomes limitations of available methods for the noise quantification across large numbers of local neighborhoods, enabling the analysis of noise differences between multiple populations and along differentiation trajectories.

We also tested simultaneous MAP inference of both $\mu_{i,j}$ and $\varepsilon_{i,j}$, and found inferred values of $\mu_{i,j}$ to be in excellent agreement with the calculated average (individual MAP estimation of $\varepsilon$, Additional file 1: Fig. S1g). However, for a small percentage of genes, simultaneous inference of $\mu_{i,j}$ and $\varepsilon_{i,j}$ results in local minima with vanishing $\mu_{i,j}$ despite a non-zero local average, supporting our robust inference strategy.

### Nuclear versus cellular transcripts exhibit elevated noise levels in peripheral blood mononuclear cells

We first applied VarID2 to test the hypothesis that nuclear export of mRNAs serves as a buffer to reduce transcriptional noise, as described for a limited set of genes measured by single-molecule fluorescent in situ hybridization (smFISH) in HeLa cells and primary Keratinocytes [23]. To compare transcriptional noise of nuclear and cytoplasmic transcripts on a genome-wide scale across a number of different cell types, we ran VarID2 on scRNA-seq and single-nucleus RNA-seq (snRNA-seq) data of human peripheral blood mononuclear cells (PBMCs, datasets generated by 10x Genomics, see Additional file 1: Table S1). For both datasets, VarID2 identified monocytes, NK cells, T cells, and B cells, which could be further sub-classified into different sub-types consistently observed in both datasets (Fig. 2a, b and Additional file 1: Fig. S2a-c). Across all cell populations, naïve T cells were found to exhibit minimal noise levels suggesting that transcriptional variability is reduced in less differentiated cell states (Fig. 2c, d). To enable the comparison of cell populations between the two datasets, we annotated corresponding cell types based on data integration with Harmony and the Seurat pipeline ("Methods" and Additional file 1: Fig. S2d). Substantial noise reduction in cellular versus nuclear transcripts was consistently observed across all cell types and for the majority of all genes (Fig. 2e and Additional file 1: Fig. S2e) independently of the expression level (Fig. 2f and Additional file 1: Fig. S2f), indicating that nuclear export could indeed facilitate noise reduction on a genome-wide level across cell types.

To validate the observation of increased noise of nuclear versus cellular transcripts, we quantified mRNA abundance by smFISH on CD8 naïve T cells isolated from human peripheral blood. We selected candidate genes with similar expression in the

nuclear and the cellular compartment, by performing differential expression analysis and keeping only genes without significant change (Additional file 1: Fig. S2f). The translational inhibitor programmed cell death-4 (*PDCD4*), involved in cell apoptosis and also in the control of CD8 T cell activation [24] exhibits increased noise in the nucleus according to our prediction (Additional file 1: Fig. S2g). Moreover, *PDCD4* undergoes alternative splicing and one of its isoforms is regulated by nuclear retention [25]. This suggests that post-transcriptional regulatory mechanisms may mediate elevated nuclear noise. In contrast, the gene encoding phosphatase inhibitor 2 (*PPP1R2*) was predicted to exhibit similar nuclear and cellular noise levels (Additional file 1: Fig. S2g). For these genes, we quantified nuclear and cytoplasmic mRNA counts by smFISH (Fig. 2g and Additional file 1: Fig. S2h), and computed the ratio of residual biological noise between nucleus and whole cells, which was consistent with the noise ratios predicted by VarID2 ("Methods" and Fig. 2h).

## Co-analysis of chromatin accessibility and gene expression noise reveals distinct modes of gene regulation

In order to gain insights into the influence of chromatin accessibility on gene expression noise, we analyzed a multiomics PBMC dataset (see Additional file 1: Table S1), which combines snRNA-seq and single-cell Assay for Transposase-Accessible Chromatin sequencing (scATAC-seq) from the same cell, by using the Signac package [26]. We focused our analysis on the chromatin accessibility across individual genes at two levels, gene activity, and individual peak signal.

Gene activity was defined as the sum of detected fragments across all peaks located in the gene body and 2 kilobases (kb) upstream of the transcriptional start site (TSS). We computed Pearson correlations for paired comparisons of expression (Ex), noise (N), and gene activity (GA) (Additional file 1: Fig. S3a). In agreement with the general notion that open chromatin promotes gene expression, we observed a substantial number of genes with a positive correlation between expression and gene activity

(See figure on next page.)
**Fig. 2** Elevated noise levels of nuclear versus whole-cell transcriptomes in human PBMCs. **a** Clustering and UMAP representation of single-nucleus RNA-seq data, consisting of human peripheral blood mononuclear cells (PBMCs) profiled with the Single Cell Multiome kit from 10x Genomics (See Additional file 1: Table S1). **b** As **a**, but showing single-cell RNA-seq data. **c** Quantification of cellular noise (average $\varepsilon$ across all genes per cell) across clusters shown in **a**. The horizontal line indicates the median of CD4 naïve T cell estimates (cluster 3), exhibiting reduced noise levels. **d** As **c**, but for cellular noise estimates for the single-cell dataset (see **b**). The horizontal line indicates the median of CD4 naïve T cell estimates (cluster 3). **e** Comparison of cellular noise levels between both datasets. The scatter plot shows the average cellular noise per cluster and their corresponding standard deviation (error bars). The x-axis corresponds to the estimates of the nucleus data and the y-axis to the cell data estimates. Similar cell populations between both datasets were identified by dataset integration, see Additional file 1: Figure S2d. **f** Gene-wise average noise in CD8 naïve T cells, comparing nucleus and cell datasets. Only genes without change in gene expression were selected and grouped into ten equally populated bins based on mean expression as shown in Additional file 1: Figure S2f. **g** Quantification of *PDCD4* (elevated nuclear noise) and *PPP1R2* (no changes in noise) expression by smFISH in human CD8 naïve T cells (see also Additional file 1: Figure S2h). Representative images of maximum intensity projections are shown. DAPI in blue, scale bar is 5 μm. **h** Noise ratio between nuclear and cellular compartments, estimated with VarID2 and smFISH. Error bars indicate standard error ("Methods"). DC, dendritic cells; NK, natural killer cells; TEM, effector memory T cells; Mono, monocytes. Boxplots in **c**, **d**, and **f**: boxes indicate inter-quartile range (IQR), and whiskers correspond to ±1.5*IQR of the box limits. Outliers beyond the whisker limits are depicted

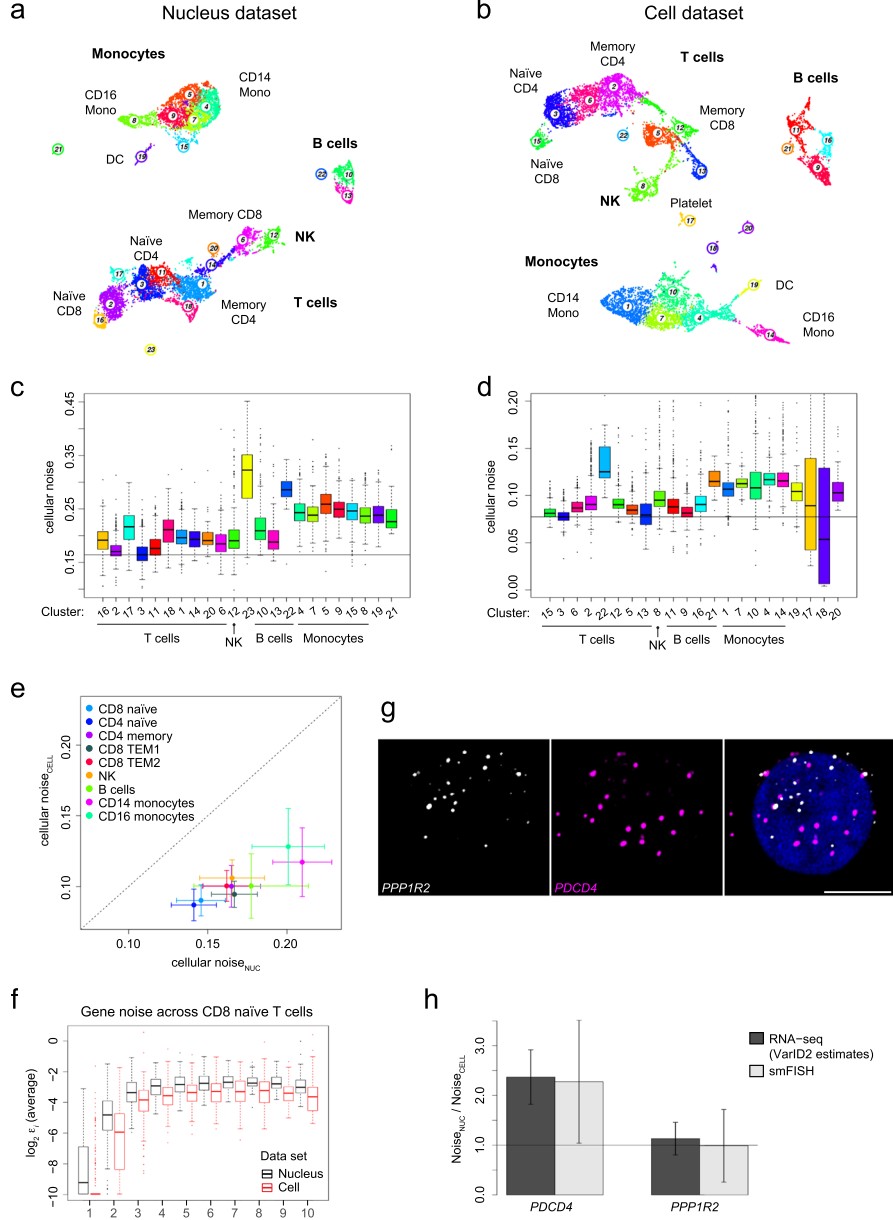

**Fig. 2** (See legend on previous page.)

(1623). However, the majority of the genes did not exhibit a clear association, potentially due to the sparsity of at least one of the modalities.

Similarly, a substantial number of genes showed a positive correlation between gene activity and noise (933, Additional file 1: Fig. S3a), and the majority of those also displayed a positive correlation of expression and gene activity (857) (termed class A genes, Fig. 3a and Additional file 1: Fig. S3b). On the other hand, most genes with a negative correlation between noise and gene activity (95) exhibited a positive expression—gene activity correlation (84) (termed class B genes, Fig. 3a and Additional file 1: Fig. S3b).

Class A genes tend to be expressed exclusively in either T cells, B cells, or monocytes, with high accessibility and noise signal in these cell types (Fig. 3b), while class

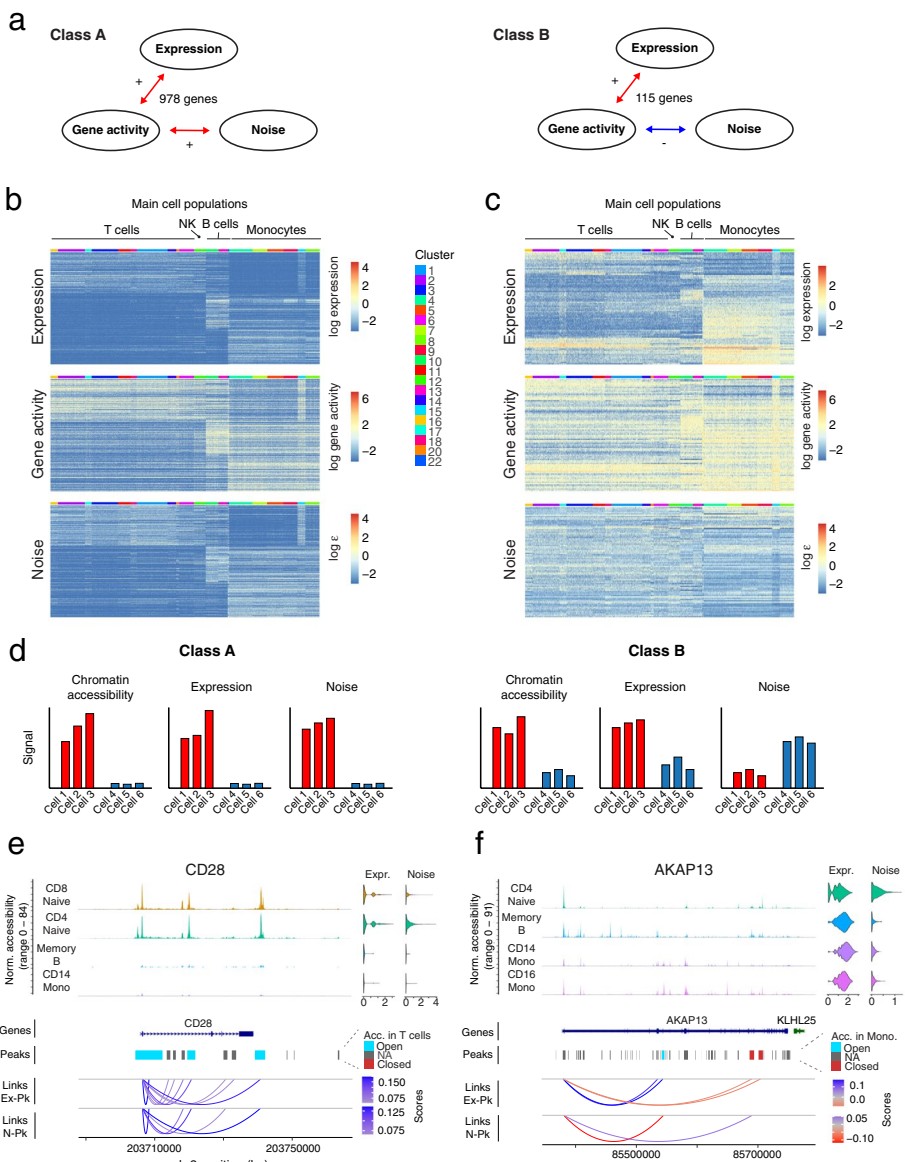

**Fig. 3** Joint analysis of chromatin accessibility, gene expression, and gene expression noise reveals gene modules with distinct modes of regulation. **a** Two sets of genes were analyzed based on the correlations in Additional file 1: Figure S3a. Class A genes (left side) have positive expression – gene activity and noise – gene activity correlations. Class B genes have positive expression – gene activity correlation but negative noise – gene activity correlation. **b** Patterns of expression (top), gene activity (middle) and noise (down) of genes belonging to class A. For convenience, a subset of ~ 300 genes is shown. **c** As **b**, but showing genes of class B. All genes in this category were included. **d** Diagram summarizing the observed patterns in chromatin accessibility, expression and noise for the set of genes in class A and class B. See main text for further details. **e** Genomic region of *CD28* (class A gene). Upper panel: normalized accessibility signal, aggregated across cells from selected clusters. Violin plots (top right) show expression and noise levels across each cluster. Differential accessibility test of T cells against the remaining dataset was performed. Peaks (middle panel) were annotated based on increased accessibility ("Open"), no change ("NA"), or decreased accessibility ("Closed"). Threshold values: $\log_2$ fold change ($\log_2$FC) > 1.25, adjusted *P* value (padj) < 0.001. Gene linkages [26] between expression and accessibility within individual peaks (links Ex-Pk) or noise and peak accessibility (links N-Pk) are shown in the lower panel, with scores corresponding to Pearson correlation coefficients. These links bind the TSS of the corresponding gene and peaks where a significant correlation was detected, and they do not represent spatial chromatin organization. **f** As **e**, but showing data of *AKAP13* (class B gene). Differential accessibility test was performed by comparing monocytes against the remaining dataset

B genes exhibit a mixture of expression patterns. While most of these genes are dominantly expressed in a specific cell population, they are still expressed at lower levels in other cell types (Fig. 3c). Other genes of class B are more ubiquitously expressed across the entire dataset. As expected, noise of class B genes is generally anti-correlated with expression. For the remaining genes (class C genes), noise and gene activity did not correlate (Additional file 1: Fig. S3c).

Hence, expression level and noise increase with chromatin accessibility for class A genes, suggesting that these genes exhibit an on–off pattern without precise control of the transcriptional level (Fig. 3d). In contrast, genes in class B show reduced variability when chromatin becomes more accessible and expression increases and may thus require more precise regulation of their transcriptional output (Fig. 3d).

Pathway enrichment analysis for these sets of genes allows to assess whether they are involved in particular cellular functions. For each main cell population (T cells, B cells, and monocytes), we performed enrichment analysis over a complete list of marker genes obtained by differential gene expression analysis ("Methods"), selecting subsets of marker genes found in class A, or those that do not belong to class A (Additional file 1: Fig. S3d). For the three cell types, marker genes belonging to class A are significantly more enriched in cell type-specific immune signaling functions compared to the full list of marker genes. Among these enriched pathways, we found, e.g., co-stimulation by the CD28 family for T cells, signaling by the B cell receptor for B cells, and interleukin 10 signaling for monocytes. In contrast, marker genes that do not belong to class A yielded more general categories in case of T cells and monocytes, and no enrichment for B cells (Additional file 1: Fig. S3d). On the other hand, enrichment analysis for marker genes within class B did not return any pathway, suggesting that only particular genes within broader functional categories require precise control of transcriptional activity.

We also searched for potential transcription factors that could regulate either class A or class B genes by performing motif enrichment analysis with RcisTarget [28]. To enrich for cell type-specific motifs, we intersected marker genes for each main cell type, i.e., T cells, monocytes, and B cells, with either class A or class B genes, and detected enriched motifs for each of these sets (Additional file 1: Fig. S3e). For instance, CEBPE, CEBPD and SPI1, which are involved in early myelomonocytic cell differentiation [29, 30], were found as potential regulators of class A monocyte marker genes. MEF2A, MEF2C, and MEF2D motifs, associated with later time points of monocyte maturation [31, 32], were enriched in class B monocyte markers. These observations suggest that class A and B genes can be controlled by different regulatory programs within the same cell type. In contrast, IRF4, a well-known regulator of T cell differentiation and activation [33], was enriched in both class A and B T cell markers.

Furthermore, we investigated associations reflected by correlations between expression or noise, respectively, and fragments at the level of individual peaks by following a recently proposed strategy [27] implemented in Signac [26]. This method addresses confounding factors such as GC content and sequence length by comparing the peak-gene correlation against a background signal and testing the significance of the correlation. We adapted the input in order to obtain both, expression – peak signal (Ex—Pk) and noise – peak signal (N—Pk) correlations. For simplicity, we focused our

analysis on peaks falling around the TSS and gene body, setting aside potential regulatory regions in *cis*.

Focusing on class A and B genes, we observed consistent patterns at the level of specific peaks as compared with gene activity signal. Genes of class A show an enrichment in both positive expression – peak and noise – peak correlations with substantial overlap of these peaks.

For instance, the T cell co-receptor *CD28* (belonging to class A) exhibits common links of positive expression – peak and noise – peak correlation (Fig. 3e). A complementary behavior was observed for the class B gene *AKAP13* which shows peaks with positive expression – peak but negative noise – peak correlation, and vice versa (Fig. 3f).

Moreover, peaks within the *CD28* locus exhibiting positive expression – peak and noise – peak correlations are differentially accessible in T cells versus other cells (Fig. 3e). Likewise, peaks associated with increased expression and low noise across the *AKAP13* sequence exhibit increased accessibility in monocytes, while peaks associated with decreased expression and high noise are less accessible (Fig. 3f). Therefore, correlations at the higher level of gene activities largely reflect the dynamics at individual peaks, supporting the distinction between class A and class B genes.

Taken together, gene expression noise can discern different modes of gene regulation corresponding to noisy on/off switches (class A) versus tight regulation of expression levels (class B).

### Gene expression noise increases during hematopoietic differentiation

To interrogate dynamics of gene expression noise during multilineage stem cell differentiation, we focused on the hematopoietic system and analyzed a dataset of $\sim 44{,}000$ mouse Kit+ hematopoietic progenitors, covering long term-HSCs (LT-HSCs), multipotent progenitors (MPPs), and fate-committed progenitors of all major blood lineages [34]. Cluster-to-cluster transition probabilities [19] predicted by VarID2 recapitulate the architecture of the hematopoietic tree (Fig. 4a and Additional file 1: Fig. S4a). LT-HSCs identified as the *Slamf1+Ly6a+Kit+Cd34*$^{\text{low}}$ *Cd48*$^{\text{low}}$ cluster 10 exhibit the lowest averaged noise level (mean noise of all genes in a local neighborhood) among all clusters (Fig. 4b and Additional file 1: Fig. S4b). Hence, transcriptional noise is suppressed in LT-HSCs, indicating a stable, transcriptionally homogenous stem cell state. For all lineages, we observed an increase of transcriptional noise with differentiation progress (Fig. 4b and Additional file 1: Fig. S4b). We further analyzed cell-to-cell transcriptome correlation within local neighborhoods (Additional file 1: Fig. S4b,c), and found that LT-HSCs are among the clusters with the highest Spearman correlation of single-cell transcriptomes. Hence, transcriptional variability in LT-HSCs is correlated across genes, suggesting fluctuations of entire gene modules.

Impaired Kit signaling affects long-term repopulation capacity of HSCs [35], and in vitro culture of W$^{41}$/W$^{41}$ mutant mice with impaired Kit kinase activity demonstrated reduced proliferation within the HSC compartment [34]. To test whether stochastic activation of cell cycle genes could underlie the perturbed exit from quiescence, we performed VarID2 analysis of scRNA-seq data generated from W$^{41}$/W$^{41}$ mutant hematopoietic progenitors [34] (Fig. 4c). We were able to identify all major hematopoietic lineages with perturbed relative abundances as reported in the original study. By matching

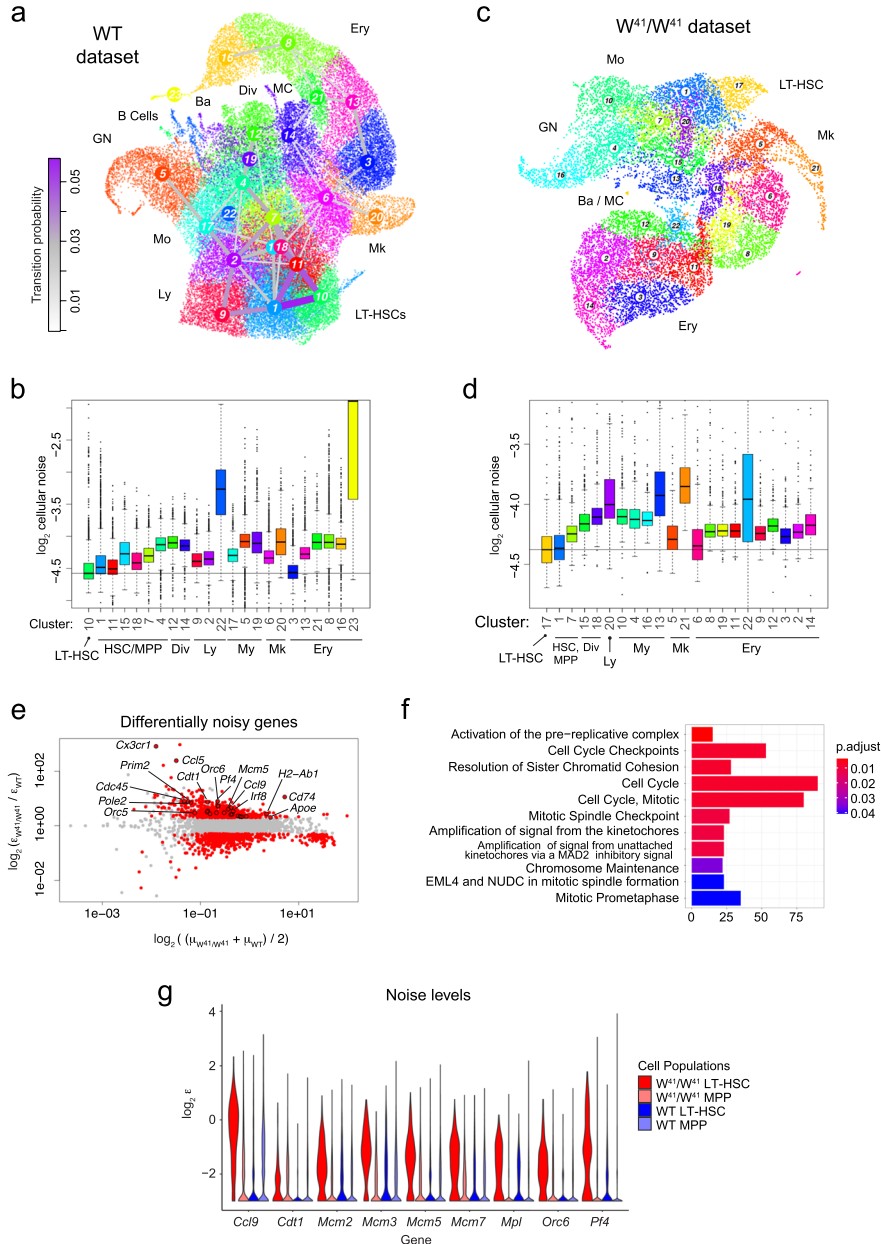

**Fig. 4** Gene expression noise increases during hematopoietic differentiation. **a** UMAP representation of hematopoietic stem and progenitor cells from the bone marrow of wildtype (WT) mice [34]. Major cell populations and VarID2 transition probabilities ("Methods") between clusters are highlighted. **b** Quantification of cellular noise (average $\varepsilon$ across all genes per cell) across clusters from the WT dataset in **a**. Horizontal line corresponds to the median noise level of the LT-HSC population. Boxes indicate inter-quartile range (IQR), and whiskers correspond to ±1.5*IQR of the box limits. Outliers beyond the whisker limits are depicted. Vertical axis limits are manually adjusted for better visualization. **c** UMAP representation of a hematopoietic stem and progenitor cells from *Kit* W[41]/W[41] mutant mice [34]. **d** As **b**, but showing cellular noise estimates of the W[41]/W[41] dataset in **c**. **e** Differentially noisy genes identified between the LT-HSC populations of W[41]/W[41] versus WT mice. MA plot shows $\log_2$FC of noise on the *y*-axis, and average expression on the *x*-axis. Threshold values: $\log_2$FC > 1, padj < 0.001. **f** Pathway enrichment analysis of the genes with increased noise in W[41]/W[41] mice from **e**. **g** Noise $\varepsilon$ estimates of genes involved in DNA replication. Quantities from each dataset were separated into LT-HSCs and the remaining cells, denoted as MPP. LT-HSC, long-term hematopoietic stem cells; MPP, multipotent progenitors; Ly, lymphocytic; My, myelocytic; Mo, monocytic; GN, granulocytic neutrophil; Ba, basophylic; MC, mast cells; Mk, megakaryocytic; Ery, erythroid; Div; dividing cells

cluster centers between wildtype and mutant datasets ("Methods"), we identified mutant cluster 17 as the unique match to the wildtype LT-HSC cluster 10 (Additional file 1: Fig. S4d), which was also supported by LT-HSC marker expression (Additional file 1: Fig. S4e). Similar to wildtype cells, mutant cells exhibit minimal noise levels in LT-HSCs and an increase upon differentiation (Fig. 4d). We next interrogated noise differences between wildtype and mutant LT-HSCs based on differentially noisy genes (Fig. 4e) and detected a strong enrichment of cell cycle genes (Fig. 4f). In particular, several members of the pre-replication complex (*Mcm2*, *Mcm3*, *Mcm5*, *Mcm7*, *Orc6*) were among the top differentially noisy genes (Fig. 4g) despite only small differences in expression levels (Additional file 1: Fig. S4f). These genes are required for the initiation of replication and showed elevated noise levels in LT-HSCs versus MPPs. Taken together, these observations suggest that cell cycle activation in $W^{41}/W^{41}$ mutant LT-HSCs becomes more stochastic. This is consistent with the observation of Dahlin et al. [34] that the number of colonies obtained from in vitro culture is overall comparable between wildtype and $W^{41}/W^{41}$ mutants, yet the frequency of very small colonies was significantly increased, indicating the presence of LT-HSCs that fail to become fully proliferative. Hence, the noise analysis can generate hypotheses consistent with the observed perturbed proliferation phenotype in *Kit* mutant mice.

### Gene expression noise increases in LT-HSCs upon ageing

Ageing increases cell-to-cell variability of CD4+T cells upon immune stimulation [36]. To test whether an increase of gene expression noise also occurs in HSCs upon ageing, and to investigate if this could explain observed phenotypic changes such as myeloid lineage bias [37], we applied VarID2 to scRNA-seq data of HSCs isolated from young (2–3 months old) and aged (17–18 months old) mice [38]. In this study, sequencing was performed in two batches (denominated as A and B) of young and aged mice, which were separated by VarID2 clusters (Fig. 5a and Additional file 1: Fig. S5a). To avoid confounding of noise quantification by batch integration, we separately analyzed clusters corresponding to the two batches. We focused our analyses on the clusters maximizing expression of LT-HSC markers (*Hlf*, *Hoxa9*, *Mecom*) within each age group and batch (Additional file 1: Fig. S5b): cluster 7, 15, 1, and 6 for young A, young B, aged A, and aged B, respectively. Compared to multipotent progenitors (MPPs), these clusters show decreased noise (Fig. 5b and Additional file 1: Fig. S5c), consistent with the analysis of data from Dahlin et al. [34]. For both batches, we observed elevated noise levels in aged versus young LT-HSCs, albeit with limited effect size for batch B (Fig. 5c), indicating that the transcriptome of LT-HSCs becomes more variable with age.

   Analysis of differentially noisy genes ("Methods" and Fig. 5d) confirmed a larger number of genes with elevated noise in aged LT-HSCs. Among these genes we detected the inhibitor of Telomerase *Terf*, cell cycle suppressers such as cyklin-dependent kinase inhibitor *Cdkn2c*, and *Gfi1b*, an essential regulator of erythro-megakaryopoiesis [39] (Fig. 5e). Furthermore, the retinoic acid-degrading enzyme *Cyb26b1*, which is required for the maintenance of dormant HSCs [40], displays elevated noise in aged LT-HSCs. Given the reduced proliferative capacity and the myeloid lineage bias of aged LT-HSCs, variability of these classes of genes could indicate the presence of differentially quiescent and lineage-biased sub-states, whereas young LT-HSCs persist in a more homogenous state.

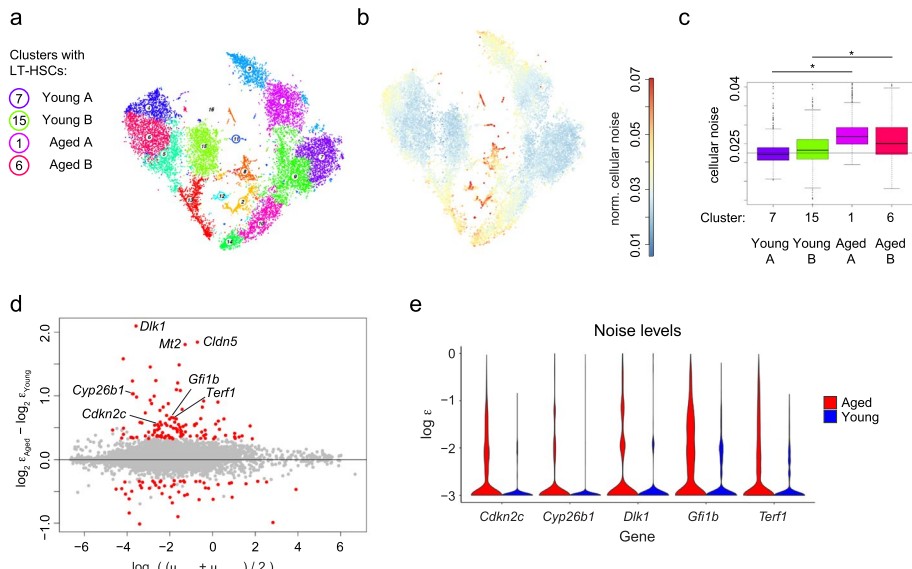

**Fig. 5** Gene expression noise increases in LT-HSCs upon ageing. **a** t-SNE representation of young and aged hematopoietic stem cells [38], sequenced in two batches A and B (see also Additional file 1: Figure S5a). LT-HSC populations identified based on marker gene expression for each condition and batch identity are highlighted (see also Additional file 1: Figure S5b). **b** t-SNE plot highlighting cellular noise estimates. **c** Comparison of cellular noise across the four LT-HSC populations identified in **a**. Boxes indicate inter-quartile range (IQR), and whiskers correspond to $\pm 1.5*$IQR of the box limits. Outliers beyond the whisker limits are depicted. Vertical axis limits are manually adjusted for better visualization. A comparison of old versus young cells for each batch was performed, $*P$ value $< 2.2e-16$ (two-sided Wilcoxon test). **d** Differentially noisy genes identified across LT-HCS populations, comparing aged versus young samples. MA plot shows $\log_2$FC of noise on the *y*-axis, and average expression on the *x*-axis. Threshold values: $\log_2$FC $> 1.25$, padj $< 0.001$. **e** Noise $\varepsilon$ estimates of some example genes detected as highly noisy in aged versus young LT-HSCs in **d**

## Dlk1 is a marker of quiescence and enhanced self-renewal of aged HSCs

To further investigate this hypothesis, we focused on *Dlk1*, the gene with the strongest noise increase in aged versus young LT-HSCs (Fig. 5d). *Dlk1* encodes a non-canonical Notch ligand which has been reported to be overexpressed in human hematopoietic CD34+ stem and progenitors from myelodysplastic syndrome patients [41]. In the ageing HSC dataset [38], *Dlk1*+ and *Dlk1*− cells intermingled in the t-SNE and did not give rise to separate clusters (Fig. 6a). Differential gene expression analysis between *Dlk1*+ and *Dlk1*− LT-HSCs ("Methods" and Fig. 6b) revealed only few differentially expressed genes such as the LT-HSC marker *Meg3* [42]. To characterize functional differences of *Dlk1*+ and *Dlk1*− LT-HSCs in more detail, we FACS-purified Dlk1+ and Dlk1− Lineage⁻Kit⁺Sca1⁺CD150⁺CD48⁻CD34⁻ HSCs from the bone marrow of aged (18 months old) mice and performed scRNA-seq by mCEL-Seq2 [43]. Gene expression analysis confirmed upregulation of *Dlk1* mRNA in sorted Dlk1+ LT-HSCs (Fig. 6c, d and Additional file 1: Fig. S6a, b). Although clustering failed to resolve *Dlk1*+ and *Dlk1*− LT-HSCs, differential gene expression analysis between sorted Dlk1+ and Dlk1− LT-HSCs further confirmed upregulation of *Meg3* and revealed significantly increased expression of the cell cycle inhibitor *Cdkn1a* and the Sulfotransferase 1A1 (*Sult1a1*) in Dlk1+ LT-HSCs (Fig. 6e). *Sult1a1* was described as a marker of the previously identified molecular overlapping (MolO) population enriched in functional HSCs obtained by four different isolation methods [44]. These observations corroborate our sorting strategy for the two

sub-populations. By FACS analysis of bone marrow cells isolated from mouse groups at different ages, we discovered that the fraction of Dlk1+ cells within the LT-HSC compartment continuously increased with age (Fig. 6f and Additional file 1: Fig. S6c, d) and positively correlated with myeloid bias (Spearman's $\rho$=0.80) in the bone marrow of ageing mice (Fig. 6g and Additional file 1: Fig. S6e).

To better understand the specific phenotype of Dlk1+ LT-HSCs, we performed a 48 h in vitro single-cell proliferation assay and observed delayed cell cycle entry compared to Dlk1− LT-HSCs (Fig. 6h). To test in vitro self-renewal of the two sub-populations, we performed a serial colony-forming unit (CFU) assay. Dlk1+ LT-HSCs exhibited significantly higher CFU capacity than Dlk1− LT-HSCs after the third re-plating (Fig. 6i), suggesting enhanced self-renewal. To assess self-renewal capacity and lineage output in vivo, we performed serial transplantations of Dlk1+, Dlk1−, or total LT-HSCs into irradiated young recipients ("Methods" and Additional file 1: Fig. S6f, g). After 16 weeks of secondary transplantations, Dlk1+ LT-HSC recipients exhibited significantly elevated chimerism in the bone marrow compared to Dlk1− LT-HSC recipients (Fig. 6j). While we also observed slightly elevated Dlk1+ chimerism in the peripheral blood (Additional file 1: Fig. S6h) and an increased HSC frequency among donor-derived cells (Additional file 1: Fig. S6i) upon secondary transplantations, the difference to Dlk1− LT-HSC recipients did not reach significance due to the large variability across individual animals. Moreover, a significantly increased myeloid lineage output was observed in the bone marrow of Dlk1+ versus Dlk1− LT-HSC recipients (Fig. 6k).

Together, the in vitro and in vivo analyses indicate a more quiescent state and increased self-renewal capacity of Dlk1+ LT-HSCs, although large variability across animals makes it difficult to quantify this effect in vivo. The significantly increased myeloid output in the bone marrow of Dlk1+ LT-HSC recipients is consistent with an intrinsic myeloid bias of these cells. Together with the observed correlation of Dlk1+ LT-HSC frequency and age-dependent myeloid bias (Fig. 6g), we conclude that expansion of Dlk1+ LT-HSCs in the bone marrow of ageing mice contributes to the known age-related myeloid bias.

(See figure on next page.)
**Fig. 6** *Dlk1* is a marker of quiescence and enhanced self-renewal in aged HSCs. **a** Expression of *Dlk1* in the dataset from Hérault et al., 2021 [38] (see Fig. 5). **b** Differentially expressed genes between *Dlk1*+ and *Dlk1*− cells across aged LT-HSCs (batch A, cluster 1 in Fig. 5a). Threshold values: $\log_2 FC > \log_2 1.25$, padj < 0.05. **c** UMAP representation of mCEL-Seq2 data of Dlk1+ and Dlk1− LT-HSC populations purified by flow cytometry. **d** As **c**, but highlighting Dlk1+ and Dlk1− LT-HSC sorted cells. **e** Differential expression analysis of the Dlk1+ versus Dlk1− sorted cells. Threshold values: $\log_2 FC > \log_2 1.25$, padj < 0.05. **f** Quantification of Dlk1+ and Dlk1− frequency among LT-HSC by flow cytometry from groups of mice with different ages (see experimental set up in Additional file 1: Figure S6c). Error bars indicate standard deviation. **g** Comparison between the percentage of Dlk1+ cells in LT-HSCs and the percentage of myeloid cells in bone marrow, corresponding to the experiment in Additional file 1: Figure S6c (see also Additional file 1: Figure S6e). Spearman's $\rho$=0.80. **h** Single-cell proliferation assay showing the number of cell divisions in LT-HSCs from young (left, 3 months old) and aged (right, 17–18 months old) mice ($n$ = 3). Error bars indicate standard deviation. **i** Serial colony-forming unit assays (CFUs) with cells isolated from aged mice (17–18 months old, $n$ = 2). Error bars indicate standard deviation. **j** Percentage of CD45.2 chimerism in bone marrow 16 weeks post transplantation, showing primary (left) and secondary (right) transplantations (see experimental set up in Additional file 1: Figure S6f). Error bars indicate standard deviation. *P* value: ns > 0.05, * ≤ 0.05 (one sided *t*-test). **k** CD42.5 lineage contribution in the bone marrow 16 weeks post transplantation, showing primary (left) and secondary (right) transplantations. Error bars indicate standard deviation. ND: non-differentiated. Statistical tests in **f**, **h**, **i**, and **k**: two-way ANOVA test; *P* value: ns > 0.05, * ≤ 0.05, ** ≤ 0.01, *** ≤ 0.001, **** ≤ 0.0001

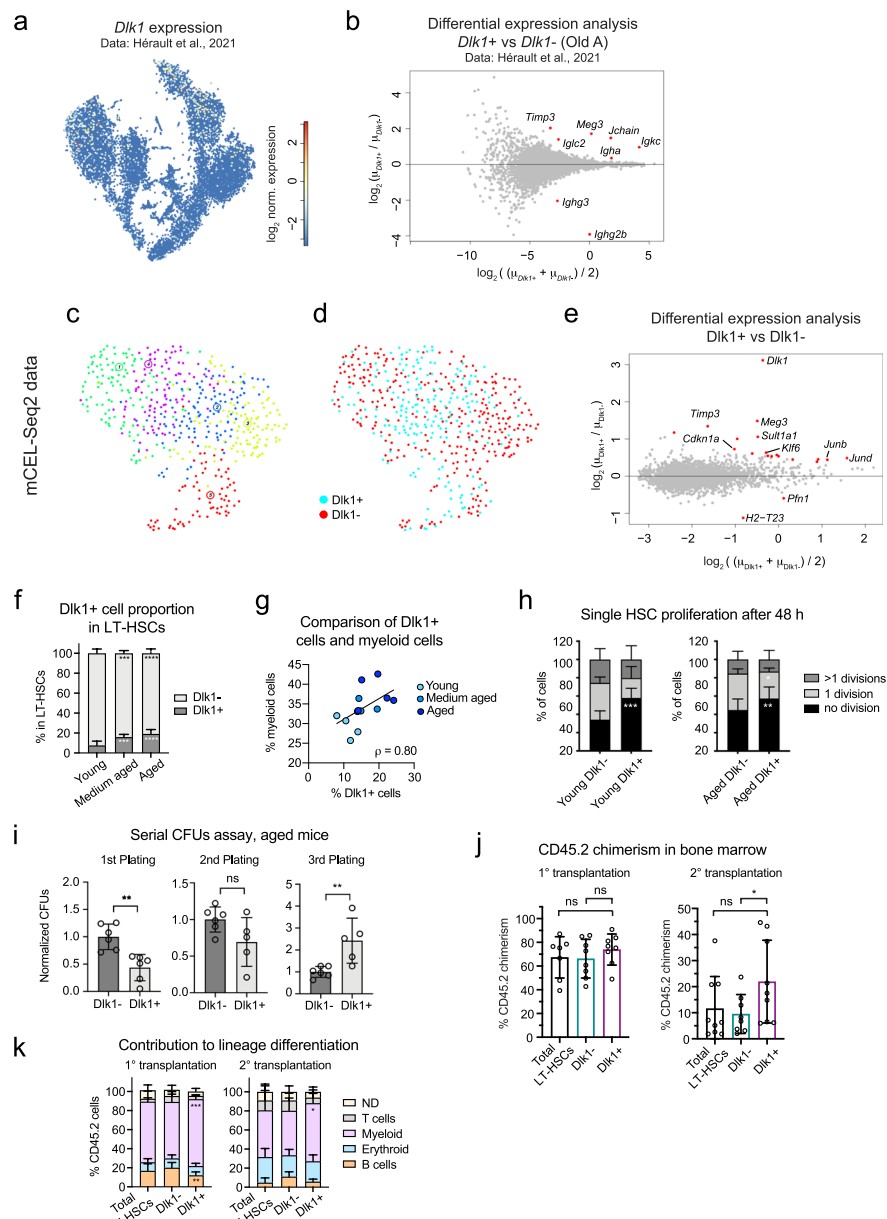

**Fig. 6** (See legend on previous page.)

Hence, differential gene expression noise has enabled the identification of a sub-type of ageing LT-HSCs with distinct functional properties, which cannot be resolved by conventional differential gene expression analysis or clustering methods.

## Discussion

VarID2 establishes a method for the quantification of local gene expression noise in cell state space. We acknowledge that the residual variability $\varepsilon$ may not be entirely free of marginal gene-specific technical noise components. However, changes in residual variability across the cell state manifold should be unaffected by such technical components,

as long as noise is independent of the mean expression. In practice, the parameter $\gamma$ driving the strength of the Cauchy prior should be adjusted such that a dependence of the average residual noise $\varepsilon$ on the total UMI count per cell is eliminated.

The ability of VarID2 to quantify biological noise across neighborhoods of tens to hundreds of thousands of cells yields unprecedented insights into the dynamics of gene expression noise along differentiation trajectories of complex multilineage systems such as bone marrow hematopoiesis. This constitutes an important angle that cannot be addressed with currently available computational methods.

Consistent with a previous study measuring increased noise levels of nuclear versus cytoplasmic mRNA for ~900 genes in HeLa cells and freshly isolated primary keratinocytes [23], our study confirms a general, genome-wide increase of biological noise in the nucleus versus the cytoplasm across multiple cell types found in human peripheral blood. Therefore, as suggested previously on a limited scale [23], nuclear export is indeed likely to confer a noise buffering function on a genome-wide level with similar effect size across diverse cell types.

Making use of scRNA-seq and scATAC-seq measurements from the same cell, we identified two classes of genes with fundamentally different noise dynamics. We hypothesize that class A genes are regulated by an on/off switch lacking precise control of expression levels, whereas transcriptional levels of class B genes need to be tightly controlled. Alternatively, a correlated increase of noise and expression of class A genes could be explained by variability of extrinsic signals, e.g., related to immune cell activation, which would be consistent with the observed enrichment of immune signaling pathways among class A genes. We speculate that particular cell type-specific functions, such as regulation of immune response, could be evolutionarily selected for to be controlled by class A genes. Such noisy switches could enable a broad spectrum of responses across a population of cells, and feedback mechanisms may lead to selection and clonal expansion of cells with the appropriate response level. In principle, the relevance of noisy responses could be experimentally investigated by replacing switch-like key regulators with more homogenously-induced alleles, e.g., by changing promoter or enhancer motifs predicted to control expression variability.

It requires further investigation to explain how these differential noise characteristics are regulated on the molecular level. We provide a starting point by demonstrating that different peaks within a given gene locus of class B genes are correlated with expression and noise, respectively. Moreover, we suggest cell-type regulators of this behavior by motif analysis, as a target for functional validation experiments.

By enabling analysis of noise dynamics during differentiation, VarID2 provides deeper insights into general properties of single-cell transcriptomes, and expands the scope of earlier work describing dynamics of transcriptome entropy during differentiation [45–47]. These studies consistently showed that stem cells maximize transcriptome, signaling, or pathway entropy compared to more differentiated states.

For the hematopoietic system, one of the best studied model systems for multilineage differentiation of stem cells, we reveal minimal noise levels in LT-HSCs, indicating that the quiescent state is transcriptionally homogenous.

However, we demonstrate that transcriptional noise in LT-HSCs increases with age whereby it always remains lower than in more differentiated progenitors, arguing for lower transcriptional fidelity and/or the emergence of transcriptionally similar sub-states of

LT-HSCs with age, which cannot be resolved by conventional clustering approaches. Our discovery of Dlk1+LT-HSCs, which exhibit higher self-renewal potential and myeloid bias than their Dlk1−counterpart, and which occur at increased frequency with age, provides evidence for the latter hypothesis. The correlation of Dlk1+LT-HSC frequency with myeloid lineage frequency in the bone marrow upon ageing, in conjunction with the cell-intrinsic myeloid bias of transplanted Dlk1+LT-HSCs, suggests this population as a determinant of age-related myeloid bias.

Due to limited transcriptional differences between Dlk1+and Dlk1−LT-HSCs, it is impossible to distinguish these populations directly by clustering and differential gene expression analysis, highlighting that gene expression noise analysis can uncover functionally distinct sub-types in seemingly homogenous cell populations.

A limitation of VarID2 is a missing link of our $\varepsilon$ estimates to parameters of a mechanistic model of transcription such as the random telegraph model of transcriptional bursting [48]. Determining transcriptional parameters such as burst size and frequency relies on the validity of the underlying assumptions of the model, which could be different from gene to gene. Moreover, in the current setting, we did not consider allele-specific quantification, which would require crossing of different genotypes and sufficient read coverage [49]. Nonetheless, the derivation of kinetic parameters of transcription represents an interesting future extension of VarID2.

## Conclusion

We here introduced VarID2, a novel method for the quantification of gene expression noise dynamics in cell state space, and demonstrate that noise dynamics are informative on fundamental properties and design principles of the transcriptome space. We showed that noise signals in stem cells can reveal the existence of functionally distinct sub-states, opening new avenues for investigating how functionally distinct cell states are molecularly encoded beyond differential gene expression, and for the elucidation of the role of transcriptional noise during cell fate decision in multilineage systems.

## Methods

### VarID2 pipeline

The original VarID method [19] was improved and extended to accommodate additional functionalities. In VarID2, homogeneous cellular neighborhoods were defined similarly as in VarID. In brief, a *k*-nearest neighbor (knn) network of cells is inferred from the Pearson residuals obtained after gene-specific normalization to eliminate the dependence on total UMI counts. VarID normalization consists of a negative binomial regression of total UMI counts akin to [50] followed by averaging regression coefficients across genes of similar expression using LOESS. In VarID2, the initial fit is performed as Poisson regression followed by a maximum likelihood inference of the dispersion parameter. As a computationally inexpensive alternative, a recently proposed analytical normalization method [51] was implemented, yielding qualitatively similar results to the full negative binomial regression. Briefly, in this normalization scheme, the regression slope coefficient equals 1 and the offset corresponds to the natural logarithm of the total UMI count, if the dependent variable is the natural logarithm of a gene's UMI count. The dispersion parameter can either be set to a fixed value, or, alternatively, inferred by

maximum likelihood. After normalization, nearest neighbors are obtained by a k-d tree search based on Euclidean distance of the Pearson residuals in a principal component analysis (PCA) reduced space. The number of principal components (PCs) to include is inferred from an elbow criterion, requiring that the difference in explained variability upon increasing the number of PCs by one is within one standard deviation across all changes upon further increasing the number of PCs (up to a maximum of 100).

The links between a central cell and each of its $k$ nearest neighbors are then tested against a negative binomial background model of UMI counts, and links to inferred outlier cells are pruned in order to obtain homogenous local cell state neighborhoods. More precisely, for every gene the hypothesis that the observed expression is explained by the respective background distribution is tested, and the $P$ value for rejecting this hypothesis is computed as the probability of residing in one of the two tails of the distribution (that is, a two-sided test is performed). The total number of null hypotheses therefore corresponds to the number of tested genes. To control for the family-wise error rate at a given $P$ value threshold, a Bonferroni correction is performed, resulting in corrected link $P$ values. The exact definition is given in the Methods section of the original VarID study [19]. In contrast to VarID, where the background distribution was inferred from a global mean–variance dependence of UMI counts across all genes, VarID2 constructs these background models locally, to better account for local variations in technical noise. Furthermore, to safeguard against false positive outliers due to sampling dropouts, a pseudocount of one is added to all UMI counts. The link probability is calculated as the geometric mean of the Bonferroni-corrected link $P$ values of the top three genes after ranking genes by link $P$ value in increasing order and adding a pseudocount of $10^{-16}$.

Furthermore, when estimating the mean expression in a local neighborhood as input to the background model, a weighted mean is computed across all neighbors with weights determined by the similarity to the central cell. The contribution of the central cell can be assigned a higher weight by the scaling parameter $\alpha$. The exact definition is given in the Methods section of the original VarID study [19]. VarID2 offers the possibility to estimate the $\alpha$ parameter, i.e., the weight of the central cell when averaging across a neighborhood for constructing the background model. The $\alpha$ parameter can be estimated in a self-consistent local way, requiring that the local average does not deviate more than one standard deviation from the actual expression in the central cell.

Clustering on the pruned knn network is performed by community detection. VarID2 offers to perform Leiden clustering [52] with adjustable resolution parameter in addition to Louvain clustering.

In VarID2 transition probabilities between two clusters are calculated as the geometric mean of the individual link probabilities connecting the two clusters (calculated as in VarID).

VarID2 implements batch correction within the negative binomial regression framework. More precisely, VarID2 utilizes a generalized linear model (GLM) for negative binomial regression of total UMI counts to eliminate the dependence of gene-specific UMI counts on the sequencing depths of a cell, akin to [50], and the GLM can be extended to include batch indicator variables to facilitate batch effect removal.

Alternatively, VarID2 has integrated Harmony for batch correction [53]. In this case, nearest neighbors are inferred post Harmony integration of the Pearson residuals resulting from the GLM-normalization.

For both batch integration strategies, pruning of integrated neighborhoods and noise inference is done based on raw UMI counts in the same way as without batch integration.

In general, we recommend avoiding batch integration and performing noise inference on individual batches, as batch integration may lead to additional sources of technical (batch) variability within local neighborhoods unaccounted for by the VarID2 technical noise model.

Finally, VarID2 facilitates pseudotime analysis along inferred lineages by integrating slingshot [54].

The central noise model has been revised as outlined in the next paragraph in order to facilitate the quantification of residual biological noise.

### VarID2 noise model

In order to quantify the actual biological variability across homogenous cellular neighborhoods, we propose a statistical model that deconvolutes defined components of variability.

The UMI count $X_{i,j}$ detected in gene $i$, $i \in \{1, \ldots, G\}$, and a central cell $j$ within a given homogeneous neighborhood follows a negative binomial distribution, with mean parameter $\mu_{i,j}$ and dispersion parameter $r_{i,j}$:

$$X_{i,j} \sim NB(\mu_{i,j}, r_{i,j}) \tag{1}$$

The variance of this distribution is given by

$$\sigma^2_{i,j} = \mu_{i,j} + \frac{1}{r_{i,j}}\mu^2_{i,j} \tag{2}$$

We used the definition of a Negative Binomial distribution as a Gamma-Poisson mixture. This way, the transcript counts $X_{i,j}$ follow a Poisson distribution with rate parameter $\lambda_{i,j}$, which in turn follows a Gamma ($\Gamma$) distribution. In our model, the rate parameter $\lambda_{i,j}$ is given by

$$\lambda_{i,j} = \beta_j \cdot \mu_{i,j} \tag{3}$$

where $\mu_{i,j}$ is the mean of the transcript counts for gene $i$ across a homogenous cellular neighborhood with a central cell $j$. A homogenous cellular neighborhood $L$ consists of the central cell $j$ and its $k$ nearest neighbors that remain after pruning: $L = \{j, j_1, j_2, \ldots, j_k\}$. $\beta_j$ is a cell-specific normalization term reflecting the local variation in total transcript counts. Variability in total UMI counts across nearest neighbor cells are caused by technical cell-to-cell variability in sequencing efficiency and by variations in cell size or RNA content [6]. We encompass all these sources of variability in a global term since we are interested in quantifying residual gene-specific variability.

Therefore, $\beta_j$ corresponds to

$$\beta_j = \frac{N_j}{\overline{N}_j} \tag{4}$$

With $N_j = \left(n_j, n_{j_1}, \ldots, n_{j_k}\right)$ representing a vector of total transcript counts $n_{j_m} = \sum_{i=1}^{G} X_{i,j_m}$ per cell $j_m$ within the neighborhood $L$ of central cell $j$. $\overline{N}_j$ is defined as the local average of total transcript counts:

$$\overline{N}_j = \frac{1}{k+1} \cdot \sum_{l \in \{j, j_1, \ldots, j_k\}} n_l \tag{5}$$

Similarly, as in [6] we propose $\beta_j$ to follow a Gamma distribution with shape parameter $\alpha_j^t$ and rate parameter $\beta_j^t$. By definition it follows for the average of $\beta_j$, $\overline{\beta}_j$, that

$$\overline{\beta}_j = 1 \tag{6}$$

and, hence,

$$\beta_j^t = \alpha_j^t \tag{7}$$

The parameter $\alpha_j^t$ is first determined by a maximum likelihood fit of a Gamma distribution to the normalized total transcript counts, which are only marginally affected by Poissonian sampling noise due to the high magnitude of these values:

$$\beta_j \sim \Gamma\left(\alpha_j^t, \alpha_j^t\right) \tag{8}$$

We infer the parameter $\alpha_j^t$ by fitting the Gamma distribution in Eq. (8) to the empirical values of $\beta_j$ in the local neighborhood of cell $j$ (see Eq. (4)). To fit a Poisson-Gamma model capturing the technical noise components defined above, and the residual biological variability, we introduce an inflation term $\varepsilon'_{i,j}$ that accounts for the biological variability of gene $i$ in the neighborhood of cell $j$.

First, in order to obtain a Gamma distribution for the Poisson rate $\lambda_{i,j}$ (see Eq. (3)), we rescale the shape and the rate parameter $\alpha_j^t$ by $1/\varepsilon'_{i,j}$, since $\alpha_j^t$ does not account for biological variability. We further multiply the rescaled rate parameter $\alpha_j^t/\varepsilon'_{i,j}$ by $1/\mu_{i,j}$ to match the mean of $\lambda_{i,j}$ which equals $\mu_{i,j}$ by definition (see Eq. (3)). This procedure yields a Gamma distribution of the Poisson rate $\lambda_{i,j}$ reflecting variability of gene-specific transcript counts:

$$\lambda_{i,j} \sim \Gamma\left(\frac{\alpha_j^t}{\varepsilon'_{i,j}}, \frac{\alpha_j^t}{\varepsilon'_{i,j} \cdot \mu_{i,j}}\right) \tag{9}$$

Second, the negative binomial distribution for transcript counts $X_{i,j}$ is determined as the corresponding Poisson-Gamma mixture

$$X_{i,j} \sim NB\left(\mu_{i,j}, r_{i,j} = \frac{\alpha_j^t}{\varepsilon'_{i,j}}\right) \tag{10}$$

with $\mu_{i,j}$ indicating the mean transcript counts per gene $i$ across local neighborhoods with central cell $j$.

The variance is given by

$$\sigma^2{}_{i,j} = \mu_{i,j} + \frac{\varepsilon'{}_{i,j}}{\alpha_j^t}\mu^2{}_{i,j} \tag{11}$$

The second term in Eq. (11) can be split into the total UMI count variability contribution $1/\alpha_j^t$ and the residual variability defined as $\varepsilon_{i,j} = (\varepsilon'{}_{i,j} - 1)/\alpha_j^t$, which scales from 0 to $\infty$. For convenience, we use $r_j^t$ instead of $\alpha_j^t$, to denote the technical dispersion parameter and rewrite the variance:

$$\sigma^2{}_{i,j} = \mu_{i,j} + \frac{1}{r_j^t} \cdot \mu^2{}_{i,j} + \varepsilon_{i,j} \cdot \mu^2{}_{i,j} \tag{12}$$

This expression encompasses the two sources of technical variability described in [6]. The mean $\mu_{i,j}$ quantifies the Poissonian noise, in which the variance scales proportionally as a function of the mean. The second technical source of variability depends on differences in sequencing efficiency or cell size and RNA content, which is captured by $r_j^t$. Therefore, we assume that the residual variability, denoted by $\varepsilon_{i,j}$, corresponds to the biological noise.

### Implementation

Since inference of the full posterior distribution by rejection sampling across all individual neighborhoods would be computationally intense, we applied Maximum a posteriori (MAP) estimation for inferring the biological variability parameter $\varepsilon_{i,j}$ that maximizes the posterior distribution:

$$\varepsilon_{i,j} = \text{argmax}_\varepsilon [ \sum_{l \in \{j,j_1,\dots,j_k\}} \log NB\left(X_{i,l}|\mu_{i,j}, r_j^t, \varepsilon\right) + \log P(\varepsilon)] \tag{13}$$

The mean expression $\mu_{i,j}$ is calculated as the arithmetic mean of UMI counts per gene $i$ across all cells $l$ within local neighborhoods $L$. $r_j^t$ is equivalent to the shape parameter $\alpha_j^t$ when fitting $\beta_j$ (defined in Eq. (4)) by a $\Gamma$ distribution.

Alternatively, we inferred both $\varepsilon_{i,j}$ and $\mu_{i,j}$ by MAP estimation, resulting in $\mu_{i,j}$ posterior estimates highly correlated to the arithmetic mean. Therefore, we omit $\mu_{i,j}$ from the optimization in order to reduce run time. Moreover, inference of $\varepsilon_{i,j}$ with fixed $\mu_{i,j}$ determined as arithmetic mean in a local neighborhood increases robustness of our inference (Additional file 1: Fig. S1g).

We propose a Cauchy distribution as a weakly informative prior distribution $P(\varepsilon)$ in order to regularize $\varepsilon_{i,j}$ posterior estimates for genes with low expression levels (Additional file 1: Fig. S1a). This prior will favor low-noise estimates in case of weak statistical support of the data, i.e., low relative likelihood.

We selected parameters of the Cauchy distribution by testing our method with a simulated dataset. In general, the location parameter $x_0$ is set to zero. For the scale parameter $\gamma$, we typically choose values in the range of 0.5 to 2, which allow to avoid noise inflation at low UMI counts. We select a value of $\gamma$ that minimizes the correlation of the average

noise and the total UMI count per cell. Technical details are provided in the vignette of the RaceID package.

### Determination of differentially noisy genes

To determine differentially noisy genes between two clusters, VarID2 applies a similar strategy as used in VarID. Briefly, a Wilcoxon rank-sum test of the noise levels in two clusters is performed. To mitigate the impact of the presence of only a small number of cells with non-zero noise estimates, a pseudocount sampled from a uniform distribution on [0:1] is added to each cell beforehand. Moreover, to account for reusing information across connected cells, the *P* value is conservatively scaled up by the number of nearest neighbors after Bonferroni correction across all genes and the final value is capped by 1.

### Data simulation

We generated a simulated data set with 34,390 genes and 100 cells, corresponding to a homogenous neighborhood. Random transcript counts were sampled from a negative binomial distribution with mean $\mu_i$ and dispersion parameter $r_i$ based on a reference dataset [22]. The mean $\mu_i$ was defined as the average of transcript counts per gene across the reference dataset and multiplied by the parameter $\alpha_j$ for cell *j*, a term accounting for the variability in sequencing efficiency or RNA content across individual cells. $\alpha_j$ was generated by sampling random values from a $\Gamma$ distribution with both shape and rate parameters equal to 2. Parameter values were selected based on the average effect size of $\alpha_j$ for real datasets.

   We estimated the total dispersion parameter per gene from the reference dataset as $r_i = \mu^2{}_i/(\sigma^2{}_i - \mu_i)$. Using this set of $r_i$ values, we took the 0.2, 0.4, and 0.8 quantiles to define three levels of additional biological noise on top of the global variability reflected by $\alpha_j$: high, medium, and low biological noise. This way, we generated a simulated dataset whose genes have three levels of variability and their expression cover the expected range (See Additional file 1: Fig. S1a).

### Gene expression variability estimation with BASiCS

We applied BASiCS [18, 20] to the simulated dataset by using the implementation with regression model and without spike-in, and choosing suggested parameters for reaching convergence: number of iterations $N = 20,000$, thinning period length Thin $= 20$; and length of burn-in period Burn $= 10,000$.

### Analysis of publicly available datasets

For analysis of public datasets, feature per barcode matrices with raw counts were retrieved from NCBI Gene Expression Omnibus (GEO; https://www.ncbi.nlm.nih.gov/geo/) or from the 10 × Genomics website (https://support.10xgenomics.com/single-cell-multiome-atac-gex/datasets). See details in Additional file 1: Table S1. We ran the VarID2 pipeline implemented in the RaceID3 package (v0.2.5) for the analysis of single-cell transcriptome data. Unless otherwise indicated, we processed the data as follows: cells with less than 1000 UMI counts were filtered out. Genes that do not

have at least 5 UMI counts in at least 5 cells were discarded. Mitochondrial genes, ribosomal genes, predicted genes with Gm-identifier and genes correlated to these classes were removed (CGenes argument in filterdata function). The pruned *k*-nearest neighbor (knn) network of cells was computed with the pruneKnn function, with number of neighbors set to 25. Clustering was performed with the Leiden algorithm for community detection [52] implemented in the graphCluster function. t-SNE or UMAP dimensional reduction representations were computed with comptsne and compumap functions with default parameters.

For local noise quantification, we adjusted the value of the prior parameter $\gamma$ based on several criteria: closeness to the simulated ground truth and reduced standard deviation of the noise estimates (See Additional file 1: Fig. S1c). We selected low values of $\gamma$ (around 0.5 and 1) in order to avoid inflation of noise estimates for lowly expressed genes. We also assessed the absence of correlation between total UMI counts and noise estimates (See Additional file 1: Fig. S4c). Cellular noise was defined as the mean noise estimates per cell, averaging across all genes.

### Human PBMCs, multiome assay, and scRNA-seq assay

*Transcriptomics data*    snRNA-seq and scRNA-seq datasets were individually analyzed with the VarID2 pipeline. To be consistent with Seurat pipelines, cells with less than 1000 or more than 25,000 UMI counts were discarded. Only mitochondrial genes, ribosomal genes and predicted genes with Gm-identifier were filtered (FGenes argument in filterdata function). To keep a comparable number of clusters between both datasets, the Leiden resolution was adjusted to 2 (nuclei data) and 1.5 (cell data). For noise estimation, we set the prior parameter $\gamma = 1$.

*Batch correction with Harmony and comparison of gene expression noise between datasets*    Matrices with raw UMI counts of snRNA-seq and scRNA-seq datasets were pooled together. Only the cells passing quality filters after VarID2 analysis were used. Batch correction with Harmony [53] was performed with the implemented function in the Seurat package [55, 56], by using default parameters and following the vignette: https://portals.broadinstitute.org/harmony/SeuratV3.html. The resulting cluster labels were used to compare cell populations, according to the following annotation: 0: CD4 memory T cells; 1, 3, 11: CD14 monocytes; 2: CD4 naïve T cells; 4: CD8 naïve T cells; 5, 7: B cells; 6: CD8 effector memory T cells 1 (TEM1); 8: natural killer cells (NK); 9: CD8 effector memory T cells 2 (TEM2); 10: CD16 monocytes.

*Noise quantification for smFISH data and comparison with sn- and scRNA-seq data*    In order to quantify noise from smFISH data, we assumed that the counts follow a negative binomial distribution with variance.

$$\sigma^2_i = \mu_i + \varepsilon^f_i \cdot \mu^2_i \tag{14}$$

where $\varepsilon^f_i$ is the reciprocal of the dispersion parameter and analogous to the noise parameter $\varepsilon_{i,j}$ obtained from our VarID2 method. Unlike scRNA-seq, smFISH is only marginally affected by technical variability of signal detection across cells. Furthermore, since we applied smFISH to naïve CD8 T cells, we do not expect substantial variation in

cell size and RNA content. Therefore, we omitted the dispersion parameter associated with total UMI count variability, which cannot be inferred based on the quantification of individual genes.

We computed the ratio of noise estimates between nuclear and cellular compartments and estimated the error based on the standard error of the mean.

*Analysis of scATAC-seq data*  scATAC-seq data was analyzed with Signac package (v1.2.1) and Seurat (v4.0.3) [55, 56]. We followed the WNN vignette of 10x Multiome, RNA + ATAC to facilitate the joint analysis of both modalities. Gene activities, defined as the sum of detected fragments across all peaks located in the gene body and 2 kb upstream of the transcriptional start site, were computed with default parameters with GeneActivity function. Peak to gene links [32] were computed with LinkPeaks function with default parameters. For expression – peak links, the Assay with expression data was used. For noise – peak links, an additional Assay denoted "Noise" was created within the Seurat object.

Differential accessibility tests were performed with FindMarkers function with the logistic regression method [57]. In brief, this method establishes a logistic model based on fragment abundance of a given feature and performs a likelihood ratio test by comparing this to a null model.

### Murine hematopoietic progenitor cells (Dahlin et al., 2018)

WT and *Kit* mutant $W^{41}/W^{41}$ samples were analyzed individually. Low-quality cells with less than 2000 UMI counts were removed. We used 50 nearest neighbors for inference of the pruned knn network. Leiden resolution: 1.5. Prior parameter $\gamma = 0.5$.

### Hematopoietic progenitors from young and aged mice (Hérault et al., 2021)

Louvain clustering was performed. t-SNE perplexity $= 200$, Prior parameter $\gamma = 0.5$.

### Transition probabilities

Transition probabilities were estimated by VarID2 [19]. Based on the connections in the pruned knn graph, probabilities of individual links connecting cells from two different clusters are estimated. Transition probabilities between two clusters correspond to the geometric mean of the individual link probabilities connecting the two clusters.

### Quadratic programming to identify similarities between two datasets

We employed quadratic programming for mapping corresponding cell populations between WT and $W^{41}/W^{41}$ samples of the HSC data [34]. We represented the cluster medoids of one dataset as a linear combination of the medoids from the other dataset. Subsequently, we optimized the weights for all cluster medoids under the constraints that they are greater or equal than zero and that they sum up to one. We solve this optimization problem with the QP function of the quadprog R package.

### Differential expression analysis

Differential expression analysis was computed with the diffexpnb function of the RaceID3 (v0.2.5) algorithm. Detection of differentially expressed genes between specific groups of cells was performed with a similar method as previously reported [58]. In brief, a negative binomial distribution which captures the gene expression variability for each group of cells if inferred based on a background model of the expected transcript count variability estimated by RaceID3 [43]. Based on the inferred distributions, a *P* value for the significance of the transcript counts between the two groups of cells is estimated and multiple testing corrected by Benjamini–Hochberg method.

### Pathway enrichment analysis

Pathway enrichment analysis was performed with the enrichPathway function from the ReactomePA R package [59] or compareCluster from clusterProfiler R package [60], with *P* value cut off = 0.05 after multiple testing correction by the Benjamini–Hochberg method. Input were ENTREZ gene IDs of genes selected by differential expression analysis or detection of differentially noisy genes.

### Experimental models and subject details

#### *Mouse*

Experiments were performed with wildtype C57BL/6 J male mice, obtained from in-house breedings or ordered from JAX. Mice were maintained under specific-pathogen-free conditions within the animal facility of the Max Planck Institute of Immunobiology and Epigenetics.

#### *Human blood samples*

Peripheral blood mononuclear cells (PBMCs) from EDTA-anticoagulated participant blood were isolated by density gradient centrifugation using Pancoll (Pan-Biotech).

### Cell suspensions and flow cytometry

Murine bone marrow (BM) cells were isolated from pooled femura, tibiae, hips, ilia, and vertebrae by gentle crushing in PBS using a mortar and pistil. Erythrocyte lysis was performed using ACK Lysing Buffer. To enrich for lineage-negative (Lin⁻) cells, Dynabeads Untouched Mouse CD4 Cells kit (Invitrogen) were used according to the manufacturer's instructions. Briefly, the erylsed BM was stained for 40 min with the provided Lineage Cocktail. Labelled cells were incubated for 15 min with polyclonal sheep anti-rat IgG-coated Dynabeads (provided in the kit). Subsequently, labelled Lin + cells were magnetically depleted. To achieve further purification, HSCs were FACS sorted. Therefore, the depleted cell fraction was stained for 30 min to 1 h using the following monoclonal antibodies: anti-lineage [anti-CD4 (clone GK1.5), anti-CD8a (53–6.7), anti-CD11b (M1/70), anti-B220 (RA3-6B2), anti-GR1 (RB6-8C5), and anti-TER119 (Ter-119)] all PE-Cy7; anti-CD117/c-Kit (2B8) in BV711; anti-Ly6a/Sca-1 (D7)-APCCy7; anti-CD34 (RAM34) in AF700; anti-CD150 (TC15-12F12.2) in PE/Dazzle; anti-CD48 (HM48-1) in BV421. Monoclonal antibodies were purchased from eBioscience, BioLegend, or MBL. Either

DLK1+ or DLK1− HSCs were sorted. Cells were sorted into Stem Pro®-34 SFM (Life Technologies) for further experiments.

### Single-cell (SiC) division assay

Single DLK1+ or DLK1− HSCs (Lineage⁻Kit⁺Sca1⁺CD150⁺CD48⁻CD34⁻) were FACS sorted into 72-well Terasaki plates and cultured in StemPro-34 SFM containing 50 ng/ml SCF, 25 ng/ml TPO, 30 ng/ml Flt3-Ligand, 100 ml/ml Penicillin/Streptomycin, and 2 mM L-Glutamine. After 48 h, each well was checked manually for the number of cell divisions under the microscope: 1 cell = no division, 2 cells = 1 division, > 2 cells = > 1 division.

### Serial colony-forming-unit assays (CFU)s

Two hundred to four hundred DLK1+ and DLK1− HSCs (Lineage⁻Kit⁺Sca1⁺CD150⁺CD48⁻CD34⁻) were FACS sorted into MethoCult M3434, plated and cultured. Approximately 7 days after the first plating, number of colonies were counted and 10,000 cells were re-plated. Second and third platings were performed 3 and 5 days, respectively, after the first re-plating. Colonies were also quantified at these time points.

### HSC transplantation assay

Six hundred Dlk1+, Dlk1−, or total HSCs (Lineage⁻Kit⁺Sca1⁺CD150⁺CD48⁻CD34⁻) isolated from 13–15-month-old CD45.2 C57BL/6 mice were transplanted into lethally irradiated (4.5 Gray + 5 Gray) CD45.1 (Ly5.1) mice together with $5 \times 10^5$ supportive spleen cells from 8–12-week-old CD45.1/2 mice within 24 h after irradiation by intravenous tail vein injection. Contribution of donor cells (CD45.2) was monitored in peripheral blood at 4, 8, 12, and 16 weeks post transplantation. For endpoint analysis, bone marrow was analyzed at 16 weeks post transplantation to quantify CD45.2 chimerism and lineage contribution. For secondary transplantations, $3 \times 10^6$ cells of whole bone marrow was isolated and retransplanted 16 weeks post transplantation.

CD45.2 chimerism and lineage contribution in bone marrow and peripheral blood were quantified by flow cytometry using the following antibodies: anti-CD45.1 (A20)– FITC, anti-CD45.2 (104)–PB, anti-CD11b (M1/70)-APCCy7, anti-GR1 (RB6.8C5)-APC, anti-CD8a (53.6.7)- PECy5, anti-CD4 (GK1.5)-PECy5, anti-B220 (RA3.6B2)-AF700.

### Amplified RNA preparation from single cells using mCEL-Seq2

The CEL-Seq2 protocol with reduced volumes was used as previously described (Herman et al., 2018) and modified using the following reagents.

Instead of 1.2 µl vapor lock as hydrophobic encapsulation barrier mineral oil (Sigma, M8410-100ML) was used. For cDNA first-strand synthesis, Protoscript II and Protoscript II Reaction Buffer (NEB, M0368L) as well as murine RNase-Inhibitor (NEB, M0314S) was used instead of SuperScript II reverse transcriptase, first-strand synthesis buffer and RnaseOUT. *Escherichia coli* DNA polymerase I, *E. coli* DNA ligase, RNase H (Invitrogen; 18,021,071) and 5 × second-strand buffer were replaced with *E. coli* DNA polymerase (NEB, M0209L), *E. coli* DNA ligase (NEB, M0205L), RNaseH (NEB, M0297S), and 10 × Second-Strand Buffer (NEB, B6117S) respectively.

The water volume was adjusted to adequately dilute the 10x second-strand buffer. After second-strand synthesis, 96 wells were pooled, which results in 96 single cells per library.

The library preparation was performed as previously described [43], but by using Protoscript II, Protoscript II Reaction Buffer, and murine RNase-Inhibitor as mentioned above instead of SuperScript II reverse transcriptase, first-strand synthesis buffer, and RnaseOUT.

### Quantification of transcript abundance

Paired-end reads were aligned to the transcriptome using BWA (version 0.6.2-r126) with default parameters [61]. The transcriptome contained all gene models based on the mouse ENCODE VM9 release downloaded from the UCSC genome browser comprising 57,207 isoforms with 57,114 isoforms mapping to fully annotated chromosomes (1–19, X, Y, M). All isoforms of the same gene were merged to a single gene locus, and gene loci were merged to larger gene groups, if loci overlapped by > 75%. This procedure resulted in 34,111 gene groups. The right mate of each read pair was mapped to the ensemble of all gene groups in the sense direction. Read mapping to multiple loci were discarded. The left mate contained the barcode information: the first six bases corresponding to the cell-specific barcode, followed by six bases representing the UMI. The remainder of the left read contained a poly(T) stretch and adjacent gene sequence. The left read was not used for quantification. For each cell barcode and gene locus, the number of UMIs was aggregated and, on the basis of binomial statistics, converted into transcript counts [6].

### smRNA FISH

Singly labelled oligonucleotides (Quasar 570 or Quasar 670) targeting *PPP1R2* and *PDCD4* mRNAs were designed with the Stellaris RNA FISH probe designer (LGC Biosearch Technologies, version 4.2) and produced by LGC Biosearch Technologies.

Naïve CD8 + T cells were isolated from the peripheral blood of two healthy donors using the Naive CD8 + T Cell Isolation kit (Miltenyi Biotec, 130–093-244) according to the manufacturer's instructions. SmRNA FISH procedure was performed in suspension, with brief centrifugations between steps (5 min at $400 \times g$) to remove the supernatant. Briefly, naïve CD8 + T cells were washed once with PBS, fixed in 3.7% formaldehyde in PBS for 10 min at room temperature, and washed again twice with PBS. Cell pellet was resuspended in 200 μl of 70% ethanol, incubated at 4 °C for 1 h and then washed with 200 μl of wash buffer A (LGC Biosearch Technologies, SMF-WA1-60) supplemented with 10% deionized formamide (Thermo Fisher Scientific, 4,440,753) at room temperature for 5 min. Cells were hybridized with 80 μl of hybridization buffer (LGC Biosearch Technologies, SMF-HB1-10) supplemented with 10% deionized formamide containing the FISH probes at a 1:100 dilution at 37 °C overnight. The next day, cells were washed with 200 μl of wash buffer A supplemented with 10% deionized formamide at 37 °C for 30 min and stained with wash buffer A supplemented with 10% deionized formamide and 10 μg/ml Hoechst 33,342 (Thermo Fisher Scientific, H3570) at 37 °C for 30 min. Cells were rinsed once with 200 μl of $2 \times SSC$, equilibrated 5 min in base glucose buffer ($2 \times SSC$, 0.4% glucose solution, 20 mM Tris pH 8.0 in RNase-free $H_2O$), and then incubated 5 min in base glucose buffer supplemented with a 1:100 dilution of glucose oxidase

(stock 3.7 mg/ml) and catalase (stock 4 mg/ml). Cell pellet was resuspended in 10 μl of ProLong Glass Antifade Mountant (Thermo Fisher Scientific, P36984) and mounted on a glass slide with a glass coverslip.

### Microscopy and image analysis

Z-stacks with 250–350-nm z-steps were acquired with the Cell Observer spinning disk confocal microscope from Zeiss with a ×100/1.40-numerical aperture oil objective lens and the PrimeBSI camera from Photometrics. Cells were segmented using Imaris image analysis software (Bitplane) and FISH spots within nucleus and cytoplasm were quantified.

## Supplementary Information

---

**Additional file 1:** Supplementary figures and tables.

**Additional file 2.** Review history

---

### Acknowledgements
We thank the Department of Medicine II of the University Hospital Freiburg, Germany for providing human blood samples.

### Review history
The review history is available as additional file 2.

### Peer review information

### Authors' contributions
DG conceived the study. RERA and DG developed the algorithm and performed all computational analyses. JR, JSH, and GD performed experiments. NCW supervised JR. DG supervised RERA, JSH, and GD. DG supervised the project. DG and RERA wrote the paper and all authors edited the paper. All authors read and approved the final manuscript.

### Funding
 DG was supported by the Max Planck Society, by the German Research Foundation (DFG) (322977937/GRK2344 MeInBio, SPP1937 GA 2129/2–2, GR4980/3–1, and SFB1425-Project #422681845), by the CZI Seed Networks for the Human Cell Atlas, and by the ERC (818846 — ImmuNiche — ERC-2018-COG). NC-W was supported by the Max Planck Society, ERC-Stg-2017 (VitASTEM; 759206), the DFG SFB1425 (Project #422681845), SFB992 (Project #192904750; B07), and CIBSS-EXC-2189 (Project ID 390939984). The funders had no role in the design of the study and collection, analysis, and interpretation of data and in writing the manuscript.

### Availability of data and materials
The list of public datasets with their references and accession numbers are in Additional file 1: Table S1, corresponding to GSE89754 [22], https://support.10xgenomics.com/single-cell-gene-expression/datasets/3.0.0/pbmc_10k_v3, https://support.10xgenomics.com/single-cell-multiome-atac-gex/datasets/1.0.0/pbmc_granulocyte_sorted_10k, GSE107727 [34] and GSE147729 [38]. The mCEL-Seq2 dataset is available in the Gene Expression Omnibus database, with accession number GSE185637 [62]. VarID2 is part of the RaceID R package (v0.2.5), available on github https://github.com/dgrun/RaceID3_StemID2_package and on CRAN. The source code for reproducing all analyses and figures is available at https://github.com/re-rosales/VarID2.git [63] under the MIT license and in Zenodo with access code https://doi.org/10.5281/zenodo.7752056 (https://zenodo.org/badge/latestdoi/616373174) [64]. The main software packages and tools used for specific analysis are enlisted in Additional file 1: Table S2.

## Declarations

### Ethics approval and consent to participate
Regarding mouse work, the protocols for animal experiments were approved by the review committee of the Max Planck Institute of Immunobiology and Epigenetics and the Regierungspräsidium Freiburg, Germany.
Related to human blood samples, these were obtained from healthy donors recruited at the Department of Medicine II of the University Hospital Freiburg, Germany. Written informed consent was given by all donors prior to blood donation.

### Competing interests
DG serves on the scientific advisory board of Gordian Biotechnology.

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

## 