## [**Additional file 2.** Review history · Genome Biology]

Review History

First round of review

Reviewer 1

Were you able to assess all statistics in the manuscript, including the appropriateness of statistical tests used? Yes: I was able to assess all statistics in the manuscript and have only one major point to highlight - that their of choice of gamma might not be universal and as such they should provide some insights into how gamma should be chosen. All other statistical tests used in the manuscript sound reasonable to me. I was not able to test the method as there are no instructions on how to do so (see comments).

Were you able to directly test the methods? No

Comments to author:

The authors present VarID2 a computational method for inferring biological noise in scRNA-seq data. By applying it on hematopoietic system, they discover that biological noise increases during hematopoietic differentiation and aging. Overall the paper is well written and the developed method will help the wider scRNA-seq community. However, I do have some comments and suggestions that will benefit the readers. I believe the method can be of broad interest to the field if the authors provide vignettes supporting their analysis.

Minor

1. Fig1a - mean-CV relationships are highlighted for two neighborhoods but no statistics are provided explaining the characteristics of the neighborhoods (median UMI/nfeatures) - it would be a helpful addition for the user as a barplot subpanel
2. GEO record for GSE185637 lists the last updated date for this dataset as May 15, 2019. Is that correct?

Major

1. The datasets used for claiming that nuclei datasets have higher noise also have 1.75 shallower sequencing depth. It is thus not possible to rule out the contribution of this difference in the estimates of epsilon. The authors also observe that ML estimates of epsilon are inflated (and control this using a Cauchy prior). I do not suspect that the difference in sequencing depth could also cause an inflation but would still recommend performing a downsampling experiment where the single-cell dataset is downsampled to have similar sequencing depth as the nuclei dataset. What then is the correlation of the epsilon estimates on the downsampled and the original dataset and does the trend (wrt. single-cell data) still hold?

2. The authors should provide some recommendations for learning γ . In the paper they move between 0.5 and 1 but it is not clear how a new user applying this tool to a new system should go about choosing one (since there is no ground truth to optimize the prior against)
3. Were the public scRNA-seq data using only the exons (i.e. mapped to the mature transcriptome and not the exon + intron RNA) - is it possible that the noise difference be attributable to difference in quantification differences?
4. Differential motif analysis in Figure 3 - what motifs are enriched differently in the two set of cells - requires a hypergeometric test or chromVAR analysis? The authors mention in the discussion ("data not shown"), but I think it is worthwhile to discuss the top hits even if they are not strong hits
5. Figure 3E (CD28) and F (AKAP13) show links that span only the gene body. Can the authors confirm that there are no distal peaks associated with these genes (within say +/- 200kb from the TSS)
6. The CRAN package is installable but I could not find instructions for running VarID2 in the vignette: <https://cran.r-project.org/web/packages/RaceID/vignettes/RaceID.html> A user can significantly benefit from more detailed vignettes - in particular discussing how γ should be chosen and steps enlisting the analyses starting from a counts matrix. This will also lead to wider user-adoption.

Reviewer 2

Were you able to assess all statistics in the manuscript, including the appropriateness of statistical tests used? Yes: My review focused on the methodological aspects.

Were you able to directly test the methods? No

Comments to author:

Rosales-Alvarez introduce VarID2, a computational approach to model transcriptional variability changes along cellular trajectories, such as those arising during development and differentiation. I believe this is a largely underexplored area of research. VarID2 builds on VarID, a related approach that was previously developed by Dr Grün's research group. The main difference between VarID2 and VarID is the ability to separate biological from technical sources of variability. This is achieved through a statistical model in which technical variability within a local neighbourhood of homogeneous cells is quantified by modelling the distribution of the total UMI counts within the neighbourhood. Residual variability that is not explained by this technical component is proposed as a metric for biological transcriptional variation. Whilst there is a number (albeit limited) of approaches that can be used to generate similar metrics of residual variability, the main advantage of VarID2 (and VarID) is its application to cellular trajectories (as opposed to being restricted to comparisons between a small number of pre-specified cell populations). As such, VarID2 may be of particular interest to those working on developmental systems.

This review focuses on the methodological aspects of the manuscript. Detailed comments are provided below.

Major comments

1. As VarID2 is an extension of a published approach, the authors have only provided a high-level overview for some of the steps implemented in VarID2 (e.g. how are the local neighbourhoods identified). Whilst this is understandable, some details require further clarification to make this manuscript more self-contained. In particular,

1a. In page 20 they authors state "the links between a central cell and its k nearest neighbours are then tested against a negative binomial background model" and such "links" are "calculated as a geometric mean of Bonferroni-corrected link probabilities". It is not clear how these "links" can be interpreted and what do the "link probabilities" represent. As the authors mention the use of Bonferroni-correction, I assume these arise from the results of a hypothesis testing approach. However, it is not clear what hypothesis is being tested.

1b. It is unclear how the weight parameter α is used when building the background model. Is it used to weight the likelihood contribution of the central cell?

2. Page 21 says that batch correction is performed using a negative binomial model but there is no explanation of how batch is incorporated in the model described in the "VarID2 noise model" section. Furthermore, if Harmony is used to perform batch correction, what is the input data used for the negative binomial noise model?

3. In the noise model, the technical noise component is quantified by modelling the distribution of total UMI counts within a local neighbourhood. How does this differ from a global scaling normalisation strategy in which the scaling factors β_j are used as fixed offsets? How would over-dispersion estimates from that model compare to the biological variability estimates generated by VarID2?

4. The definition of the Gamma-Poisson mixture is clear as combining equations (1), (3) and (8) leads to a Negative Binomial distribution with mean equal to μ_{ij} and dispersion equal to α_j^t . What is unclear is: why does this imply that the distribution for λ_i (λ_i is not defined, should this be λ_{ij} ?) corresponds to equation (10)? Moreover, is the proposed model based on a different formulation for λ_{ij} ? If the decomposition is $\lambda_{ij} = \lambda_i \mu_{ij}$, then there is a missing N_j term in equation (10) as the expected value of λ_i is not equal to one. Please clarify.

5. Page 25 introduces a weakly informative Cauchy prior used to regularise posterior estimates for ϵ , removing the inflation that was observed for lowly expressed genes. When comparing ML and MAP estimates in Fig 1e, I can see that the use of a Cauchy prior "flips" the sign of the estimation bias for lowly expressed genes. However, as seen in Fig 1f, a systematic bias remains and ϵ tends to be under-estimated for lowly expressed genes. More generally, ϵ estimates are noisier for lowly expressed genes and therefore VarID2 results are less reliable for such genes (this noise would likely lead to a higher variance in the posterior distribution and this higher uncertainty could prevent downstream effects, but MAP estimates will not capture that). This noise is not unique to VarID2 and indeed a similar issue arises when applying BASiCS (Fig S1f). Using their simulations, the authors could suggest an expression threshold below which ϵ are unreliable. This would be very helpful for potential users and would guide gene-level QC steps that need to be applied (I note that the simulations include over 30K genes, suggesting that no gene-level QC is currently suggested).

6. Figs S1d and S1e . The comparison between VarID2 and BASiCS is affected by the issue highlighted above: estimates generated by both approaches are highly correlated for highly expressed genes and the biggest discrepancies are observed in the low end of expression, where both methods are less reliable. Applying a gene-level QC step will therefore significantly increase the correlation between both approaches and provide a more meaningful comparison.

7. Regarding the comparison between VarID2 and BASiCS. Both approaches are based on a similar Poisson-Gamma mixture and it is reassuring to see that transcriptional variability estimates generated by both methods are highly correlated. However, a direct comparison may be misleading as they are on a different scale. Indeed, VarID2 defines this value in terms of a factor that multiplies μ_{ij} in the variance decomposition (equation (12)). Instead, BASiCS defines residual variability estimates as deviations with respect to an overall trend, which is calculated in a logarithmic scale of the Negative Binomial dispersion parameter. This means that the biases reported in Fig S1f are not accurate as both methods are estimating different quantities.

8. It is great to see that both the data and software are publicly available. I would encourage the authors to also publish their analysis code as that would improve the reproducibility aspects of their work (for example, some software may require parameter inputs that are missing in the main text).

Minor comments

1. Perhaps you can simplify to use β_j instead of $\llbracket \beta' \rrbracket_j$? That would simplify the notation (it is unclear why the ' is needed)

2. The authors use equation (6) to justify why a single parameter is used to model the shape and rate of the Gamma distribution. Whilst this provides a good intuition for this choice, the formal interpretation is a bit more nuisance. In particular, equation (6) refers to the sample average for cell-specific factors that are already inferred using the available data. In that context, the average is indeed exactly equal to 1. However, equation (8) means that the expected value for $\llbracket \beta' \rrbracket_j$ is equal to 1.

3. Page 20 states that a pseudo-count (=1) was added to safeguard against false positive due to outliers. However, as noted by Townes et al (2019), the addition of a pseudo-count may introduce a larger gap between zero and non-zero observations. In such cases, adding a pseudo-count could lead to more extreme outliers. It would be good to discuss how the observation by Townes et al (2019) may affect VarID2 and whether the results are robust to different pseudo-count values.

4. Scatter-plots visualising several thousands of dots (such as Fig 1e) can be misleading unless the density of dots is incorporated.

5. Version numbers are missing for some software packages. The number of iterations is also missing for BASiCS.

References

Townes et al (2019) <https://doi.org/10.1186/s13059-019-1861-6>

We thank the reviewers for their constructive criticism. Our responses to the reviewers are highlighted in blue and *italics*.

Reviewer #1:

The authors present VarID2 a computational method for inferring biological noise in scRNA-seq data. By applying it on hematopoietic system, they discover that biological noise increases during hematopoietic differentiation and aging. Overall the paper is well written and the developed method will help the wider scRNA-seq community. However, I do have some comments and suggestions that will benefit the readers. I believe the method can be of broad interest to the field if the authors provide vignettes supporting their analysis.

We thank the reviewer for the positive evaluation of our work and appreciate the comments and suggestions, which helped us to improve our manuscript.

Minor

1. Fig1a - mean-CV relationships are highlighted for two neighborhoods but no statistics are provided explaining the characteristics of the neighborhoods (median UMI/nfeatures) - it would be a helpful addition for the user as a barplot subpanel

We agree that this would be useful information and added a subpanel of violin plots to Fig. 1a showing the distribution of UMI and feature counts for the two neighborhoods.

2. GEO record for GSE185637 lists the last updated date for this dataset as May 15, 2019. Is that correct?

We are not sure, what went wrong, but the dataset was uploaded in October 2021:

<https://www.ncbi.nlm.nih.gov/geo/query/acc.cgi?acc=GSE185637>

*We provide access to this record with the following reviewer token: **ivapggguuvbexjud***

Major

1. The datasets used for claiming that nuclei datasets have higher noise also have 1.75 shallower sequencing depth. It is thus not possible to rule out the contribution of this difference in the estimates of epsilon. The authors also observe that ML estimates of epsilon are inflated (and control this using a Cauchy prior). I do not suspect that the difference in sequencing depth could also cause an inflation but would still recommend performing a downsampling experiment where the single-cell dataset is downsampled to have similar sequencing depth as the nuclei dataset. What then is the correlation of the epsilon estimates on the downsampled and the original dataset and does the trend (wrt. single-cell data) still hold?

We agree with the reviewer that differences in sequencing depth could potentially affect the comparison of noise estimates for nuclei and whole cells. We followed the recommendation of the reviewer and performed a down-sampling analysis. For this, we tried to replicate the UMI count distribution of the nuclei data by defining the average proportion of nuclei-to-cell UMIs and randomly sampling these proportion of counts from the cell data. Although noise estimates derived for down-sampled single-cell RNA-seq were slightly reduced compared to the original data (Reviewer Fig. 1a-b), as a consequence of reduced statistical power to detect residual biological noise at low UMI counts, noise differences between single-cell and single-nuclei data were only marginally affected by down-sampling, supporting the robustness of our biological noise inference with regard to variations in sequencing depth (Reviewer Fig. 1c).

Reviewer Fig. 1: Comparison of noise estimates across single-nucleus, single-cell (original) and down-sampled single-cell datasets. **a** Cellular noise across the main cell populations for the three datasets. **b** Comparison of cellular noise levels between nuclei and down-sampled cell datasets. The dot plot shows the average cellular noise per cluster and the corresponding standard deviation (error bars). **c** Correlation of cellular noise levels between the original (x-axis) and the down-sampled (y-axis) cell datasets. Continuous line: $y=x$; dotted lines: $\pm \log$ fold change. Pearson's correlation coefficient (R) of log-transformed data is indicated.

2. The authors should provide some recommendations for learning γ . In the paper they move between 0.5 and 1 but it is not clear how a new user applying this tool to a new system should go about choosing one (since there is no ground truth to optimize the prior against)

For selection of the prior parameter γ we search for values that avoid noise inflation at low UMI counts. These values are typically between 0.5 to 2 for our simulated dataset (Fig. S1a,c). To determine an appropriate value, we minimize the correlation between total UMI counts and average noise ε . In the vignette of the RaceID/VarID2 package we provide more detailed explanation of how this is practically done. We provided a function (plotUMINoise) to inspect the dependence between average ε and total UMI counts per cell.

In the revised manuscript we added more extensive explanations to the Methods section:

“For the scale parameter γ we typically choose values in the range of 0.5 to 2, which allow to avoid noise inflation at low UMI counts. We select a value of γ that minimizes the correlation of the average noise and the total UMI count per cell. Technical details are provided in the vignette of the RaceID package.”

3. Were the public scRNA-seq data using only the exons (i.e. mapped to the mature transcriptome and not the exon + intron RNA) - is it possible that the noise difference be attributable to difference in quantification differences?

We thank the reviewer for raising this valid point. In the comparison of nuclear and cellular transcripts, the single-cell RNA-seq data were mapped only to exonic regions, while the single-nuclei RNA-seq data were mapped to unprocessed transcripts comprising exonic and intronic regions. To follow the recommendation of the reviewer, we computed a new alignment of the cell dataset to primary transcripts comprising both exonic and intronic regions. Since this strategy substantially increased the UMI count per gene, exacerbating the difference between single-cell and single-nucleus RNA-seq data, we down-sampled the newly mapped single-cell RNA-seq data in order to obtain comparable UMI count distributions for both datasets. Consistent with the analysis presented in the manuscript, we observed elevated noise levels for transcript in the nucleus versus whole cell (Reviewer Fig. 2).

Reviewer Fig. 2: Comparison of noise estimates across nucleus and cell (down-sampled) datasets, both mapped to primary transcripts comprising exonic and intronic regions. Cell data were down-sampled in order to obtain comparable UMI count distributions for both datasets. Left: Violin plot comparing noise in the nucleus and the whole cell across the main cell

populations. Right: Comparison of noise levels between nuclei and cell (down-sampled) datasets. The dots indicate average noise per cluster and their corresponding standard deviation (error bars).

We note that differences in library size can affect noise estimates as a consequence of reduced statistical power at low UMI count. For this reason, we compared genes with similar expression levels within CD8 naïve T cells as an example (Fig. 2f) in the main manuscripts.

We would like to point out the interpretation of biological noise estimates is affected by the mapping strategy. While noise estimates derived from exon-mapping of single-cell RNA-seq data primarily affect noise of processed, mature mRNAs, noise estimates inferred from data mapped to exonic and intronic regions reflect a convolution of noise in mature and unprocessed/incompletely processed transcripts. Since we would like to compare noise estimates of mature mRNAs in the cell to unprocessed mRNA in the nucleus, we prefer to present the data obtained by our original mapping approach in the manuscript.

4. Differential motif analysis in Figure 3 - what motifs are enriched differently in the two set of cells - requires a hypergeometric test or chromVAR analysis? The authors mention in the discussion ("data not shown"), but I think it is worthwhile to discuss the top hits even if they are not strong hits.

We agree with the reviewer that this information is of interest. We extended our original analysis by running RcisTarget, which includes a broad database of motifs (Aibar et al., 2017). To enrich for cell-specific motifs, we performed an overlap of highly expressed genes in one of the main cell types (T cells, monocytes and B cells) and either class A or class B genes. The three motifs with highest enrichment per category are shown in Reviewer Fig. 3.

Reviewer Fig. 3: Motif enrichment analysis performed for genes with increased expression in one of the main PBMC cell types, i.e., T cells, Monocytes or B cells, that belong to either class A or class B genes (as defined in the manuscript). The top three motifs with highest enrichment within each category are shown, and the normalized enrichment score (NES) is indicated.

This plot was added into Fig. S3e. We also included an explanation into the results section of the manuscript:

“We also searched for potential transcription factors that could regulate either class A or class B genes by performing motif enrichment analysis with RcisTarget [27]. To enrich for cell type-specific motifs, we intersected marker genes for each main cell type, i.e., T cells, monocytes and B cells, with either class A or class B genes, and detected enriched motifs for each of these sets (Fig S3e). For instance, CEBPE, CEBPD and SPI1, which are involved in early myelomonocytic cell differentiation [28], were found as potential regulators of class A monocyte marker genes. MEF2A, MEF2C and MEF2D motifs, associated with later time points of monocyte maturation [29,30], were enriched in class B monocyte markers. These observations suggest that class A and B genes can be controlled by different regulatory programs within the same cell type. In contrast, IRF4, a well-known regulator of T cell differentiation and activation [31], was enriched in both class A- and B T cell markers.”

We also adjusted the discussion section:

“Moreover, we suggest cell-type regulators of this behavior by motif analysis, which are subject to further validation.”

5. Figure 3E (CD28) and F (AKAP13) show links that span only the gene body. Can the authors confirm that there are no distal peaks associated with these genes (within say +/-200kb from the TSS)

In the manuscript, we tried to include a wide region by extending the genomic tracks 15,000 bp upstream and 25,000 bp downstream from the TSS. In fact, there are very few links that go beyond these distances. However, the signal on the genomic tracks is less intense and difficult to distinguish. We provide an extended version of the genomic tracks in Reviewer Fig. 4 and 5, covering the genomic regions in which all links were detected. Since the additional information from these extended tracks is limited we would prefer to keep the original plots in the manuscript.

Reviewer Fig. 4: Genomic region of CD28 (class A gene), expanding the regions with detected links (500kb up- and downstream from the TSS). Upper panel: normalized accessibility signal, aggregated across cells from selected clusters. Violin plots (top right) show expression and noise levels for each cluster. Middle panel: Upper row shows all detected peaks. Bottom row shows peaks with differential accessibility and highlights peaks as “open” or “closed” based on a differential accessibility test of T cells against the remaining dataset. Threshold values: $\log_{FC} > \log(1.25)$, adjusted $P < 0.001$. Gene linkages [26,32] between expression and accessibility within individual peaks (links Ex-Pk) or noise and peak accessibility (links N-Pk) are shown in the lower panel, with scores corresponding to Pearson correlation coefficients. These links bind the TSS of the corresponding gene and peaks where a significant correlation was detected, and do not represent spatial chromatin organization.

Reviewer Fig. 5: Genomic region of *AKAP13* (class B gene), expanding the regions with detected links (380kb upstream and 50kb downstream from the TSS). Similar to **Reviewer Fig. 4**. Differential accessibility test was performed by comparing monocytes against the remaining dataset.

6. The CRAN package is installable but I could not find instructions for running *VarID2* in the vignette: <https://cran.r-project.org/web/packages/RaceID/vignettes/RaceID.html> A user can significantly benefit from more detailed vignettes - in particular discussing how gamma should be chosen and steps enlisting the analyses starting from a counts matrix. This will also lead to wider user-adoption.

We thank the reviewer for bringing the lack of clarity to our attention. In fact, the vignette already contained detailed instructions for running VarID2. However, the method was introduced as VarID in the previous version of the vignette. We now make it clear in the vignette that VarID2 replaces the previous VarID, since it comes with additional functionalities (the original VarID functions remain available in the package), and replace VarID by VarID2 where appropriate. The vignette contains a detailed step-by-step explanation of how to run VarID2 starting from a raw count matrix for a dataset provided in the package (intestinalData). See section "VarID2". In particular, we included instructions on how to choose the gamma prior parameter. Please refer to the explanation of the plotUMINoise function,

which allows plotting mean noise estimates per cell as a function of average UMI count. Gamma should be chosen such that the correlation is minimized. The updated package RaceID v0.3.0 has been uploaded to github and was submitted to CRAN.

Reviewer #2:

Rosales-Alvarez introduce VarID2, a computational approach to model transcriptional variability changes along cellular trajectories, such as those arising during development and differentiation. I believe this is a largely underexplored area of research. VarID2 builds on VarID, a related approach that was previously developed by Dr Grün's research group. The main difference between VarID2 and VarID is the ability to separate biological from technical sources of variability. This is achieved through a statistical model in which technical variability within a local neighbourhood of homogeneous cells is quantified by modelling the distribution of the total UMI counts within the neighbourhood. Residual variability that is not explained by this technical component is proposed as a metric for biological transcriptional variation. Whilst there is a number (albeit limited) of approaches that can be used to generate similar metrics of residual variability, the main advantage of VarID2 (and VarID) is its application to cellular trajectories (as opposed to being restricted to comparisons between a small number of pre-specified cell populations). As such, VarID2 may be of particular interest to those working on developmental systems.

We thank the reviewer for acknowledging the novelty of our VarID2 method, and we appreciate the constructive feedback that enabled us to improve the manuscript.

This review focuses on the methodological aspects of the manuscript. Detailed comments are provided below.

Major comments

1. As VarID2 is an extension of a published approach, the authors have only provided a high-level overview for some of the steps implemented in VarID2 (e.g. how are the local neighbourhoods identified). Whilst this is understandable, some details require further clarification to make this manuscript more self-contained. In particular,

1a. In page 20 they authors state "the links between a central cell and its k nearest neighbours are then tested against a negative binomial background model" and such "links" are "calculated as a geometric mean of Bonferroni-corrected link probabilities". It is not clear how these "links" can be interpreted and what do the "link probabilities" represent. As the authors mention the use of Bonferroni-correction, I assume these arise from the results of a hypothesis testing approach. However, it is not clear what hypothesis is being tested.

1b. It is unclear how the weight parameter α is used when building the background model. Is it used to weight the likelihood contribution of the central cell?

We acknowledge that it is important to provide sufficient explanations in the Methods section to make the manuscript more self-contained. We therefore expanded the Methods section to provide a more detailed description (revised text is highlighted in bold):

*"The links between a central cell and each of its k nearest neighbors are then tested against a negative binomial background model of UMI counts, and links to inferred outlier cells are pruned in order to obtain homogenous local cell state neighborhoods. **More precisely, for every gene the hypothesis that the observed expression is explained by the respective background distribution was tested, and the P value for rejecting this hypothesis was***

computed as the probability of residing in one of the two tails of the distribution (that is, a two-sided test is performed). The total number of null hypotheses therefore corresponds to the number of tested genes. To control for the family-wise error rate at a given P value threshold, a Bonferroni correction was performed, resulting in corrected link P values. The exact definition is given in the Methods section of the original VarID manuscript [19]. In contrast to VarID, where the background distribution was inferred from a global mean-variance dependence of UMI counts across all genes, VarID2 constructs these background models locally, to better account for local variations in technical noise. Furthermore, to safeguard against false positive outliers due to sampling dropouts, a pseudocount of one was added to all UMI counts. The link probability is calculated as the geometric mean of the Bonferroni-corrected link P values of the top three genes after ranking genes by link P value in increasing order and adding a pseudocount of 10^{-16} .

Furthermore, when estimating the mean expression in a local neighborhood as input to the background model, a weighted mean is computed across all neighbors with weights determined by the similarity to the central cell. The contribution of the central cell can be assigned a higher weight by the scaling parameter α . The exact definition is given in the Methods section of the original VarID manuscript [19]. VarID2 offers the possibility to estimate the α parameter, i.e., the weight of the central cell when averaging across a neighborhood for constructing the background model. The α parameter can be estimated in a self-consistent local way, requiring that the local average does not deviate more than one standard deviation from the actual expression in the central cell.”

2. Page 21 says that batch correction is performed using a negative binomial model but there is no explanation of how batch is incorporated in the model described in the "VarID2 noise model" section. Furthermore, if Harmony is used to perform batch correction, what is the input data used for the negative binomial noise model?

We thank the reviewer for bringing up this lack of clarity, and we now rewrote this paragraph to provide more detailed explanations:

“VarID2 implements batch correction within the negative binomial regression framework. More precisely, VarID2 utilizes a generalized linear model (GLM) for negative binomial regression of total UMI counts to eliminate the dependence of gene-specific UMI counts on the sequencing depths of a cell, akin to [49], and the GLM can be extended to include batch indicator variables to facilitate batch effect removal.

Alternatively, VarID2 has integrated Harmony for batch correction [52]. In this case, nearest neighbors are inferred post Harmony integration of the Pearson’s residuals resulting from the GLM-normalization.

For both batch integration strategies, pruning of integrated neighborhoods is done based on raw UMI counts in the same way as without batch integration.

In general, we recommend avoiding batch integration and performing noise inference on individual batches, as batch integration may lead to additional sources of technical (batch) variability within local neighborhoods unaccounted for by the VarID2 technical noise model.”

3. In the noise model, the technical noise component is quantified by modelling the distribution of total UMI counts within a local neighbourhood. How does this differ from a

global scaling normalisation strategy in which the scaling factors β'_j are used as fixed offsets? How would over-dispersion estimates from that model compare to the biological variability estimates generated by VarID2?

A global scaling normalization by neighborhood-specific scaling factors would not allow the deconvolution of noise components anymore. For example, rescaling cells within a neighborhood by multiplication with $1/\beta'_j$, which would correspond to a common size-factor normalization, would inflate sampling noise for cells with low UMI counts while dampening sampling noise for cells with very high UMI counts. Hence, deconvolution of technical and biological noise in local neighborhoods would not be possible for normalized data. We show this effect in Reviewer Fig. 6.

Reviewer Fig. 6: CV as a function of mean on a logarithmic scale for raw UMI counts (left) and after size factor normalization (right). Sample data of murine bone marrow cells from Tusi et al. [22].

4. The definition of the Gamma-Poisson mixture is clear as combining equations (1), (3) and (8) leads to a Negative Binomial distribution with mean equal to μ_{ij} and dispersion equal to α_j^t . What is unclear is: why does this imply that the distribution for λ_i (λ_i is not defined, should this be λ_{ij} ?) corresponds to equation (10)? Moreover, is the proposed model based on a different formulation for λ_{ij} ? If the decomposition is $\lambda_{ij} = \lambda_i \mu_{ij}$, then there is a missing N_j term in equation (10) as the expected value of λ_i is not equal to one. Please clarify.

We apologize for the lack of clarity, and we also discovered a typo in equation (9). Instead of N_j it should be μ_{ij} in the denominator. Furthermore, the reviewer is correct in pointing out that it is λ_{ij} instead of λ_i . We corrected this in the revised version.

In order to model the effect of technical noise captured by β'_j on the UMI count distribution of gene i , we multiply the rate parameter of the Gamma distribution by $1/\mu_{ij}$ which rescales the mean of the distribution to μ_{ij} . The corresponding Gamma-Poisson mixture then gives rise to the Negative Binomial distribution in equation (10).

We provide more detailed explanations in the revised manuscript:

“We further multiply the rate parameter of the Gamma distribution by $1/\mu_{ij}$ to rescale the mean of the distribution to μ_{ij} in order to obtain a distribution of the Poisson rate λ_{ij} reflecting variability of gene-specific transcript counts.”

5. Page 25 introduces a weakly informative Cauchy prior used to regularise posterior estimates for ϵ , removing the inflation that was observed for lowly expressed genes. When comparing ML and MAP estimates in Fig 1e, I can see that the use of a Cauchy prior "flips" the sign of the estimation bias for lowly expressed genes. However, as seen in Fig 1f, a systematic bias remains and ϵ tends to be under-estimated for lowly expressed genes. More generally, ϵ estimates are noisier for lowly expressed genes and therefore VarID2 results are less reliable for such genes (this noise would likely lead to a higher variance in the posterior distribution and this higher uncertainty could prevent downstream effects, but MAP estimates will not capture that). This noise is not unique to VarID2 and indeed a similar issue arises when applying BASiCS (Fig S1f). Using their simulations, the authors could suggest an expression threshold below which ϵ are unreliable. This would be very helpful for potential users and would guide gene-level QC steps that need to be applied (I note that the simulations include over 30K genes, suggesting that no gene-level QC is currently suggested).

We agree with this observation described by the reviewer and appreciate the suggestion. We explored the idea of removing lowly expressed genes by testing different mean expression thresholds. Using our simulated dataset, we aimed at maximizing the proportion of remaining genes after filtering and the proportion of genes with ϵ estimates close to the ground truth (within a \log_2 -fold change of one around the ground truth). However, we would like to propose filtering as optional, since relevant genes with low expression may be lost from the dataset.

We included a detailed explanation into the Results section:

"Notwithstanding, noise estimates for lowly expressed genes tend to deviate from the ground truth due to limited statistical power. In order to filter out lowly expressed genes, we tested different thresholds of gene expression, and we assessed if ϵ estimates were within a 2-fold confidence interval around the ground truth. We suggest an optional expression threshold between 0.3 and 0.4, since it preserves around 60% of genes within the dataset and 60% to 90% of those have noise estimates within the ground truth confidence interval (Fig. S1d)."

Moreover, we added an additional panel as Fig. S1d and adjusted the rest of the panels accordingly.

6. Figs S1d and S1e . The comparison between VarID2 and BASiCS is affected by the issue highlighted above: estimates generated by both approaches are highly correlated for highly expressed genes and the biggest discrepancies are observed in the low end of expression, where both methods are less reliable. Applying a gene-level QC step will therefore significantly increase the correlation between both approaches and provide a more meaningful comparison.

We applied the QC criterion as explained in our response to the previous point. For the comparison with BASiCS, we show in Reviewer Fig. 7 that the correlation between VarID2 estimates (ϵ_{MAP}) and BASiCS estimates (ϵ_{BASiCS}) increases slightly when filtering out genes with mean expression $\mu < 0.4$. A more pronounced increase in the correlation coefficient was

observed when comparing ML (maximum likelihood estimation) and BASiCS, which can be explained by unreliable noise estimates for these lowly expressed genes if regularization by our prior is dropped.

Reviewer Fig. 7: Left: comparison of ϵ_{MAP} estimates (VarID2 noise estimates) with ϵ_{BASiCS} (the dispersion parameter ϵ computed by BASiCS). Right: maximum likelihood estimates ϵ_{ML} and ϵ_{BASiCS} estimates. In both plots, noise levels of the simulated dataset are highlighted as well as the genes that are removed by filtering out genes with mean expression $\mu < 0.4$.

7. Regarding the comparison between VarID2 and BASiCS. Both approaches are based on a similar Poisson-Gamma mixture and it is reassuring to see that transcriptional variability estimates generated by both methods are highly correlated. However, a direct comparison may be misleading as they are on a different scale. Indeed, VarID2 defines this value in terms of a factor that multiplies μ_{ij} in the variance decomposition (equation (12)). Instead, BASiCS defines residual variability estimates as deviations with respect to an overall trend, which is calculated in a logarithmic scale of the Negative Binomial dispersion parameter. This means that the biases reported in Fig S1f are not accurate as both methods are estimating different quantities.

We thank the reviewer for pointing out that the direct comparison of the BASiCS-derived parameter with the simulated ground truths variability is problematic since the BASiCS parameters is defined in a different way. We agree with the reviewer and removed Fig. S1f and the corresponding statement in the results section from the manuscript.

8. It is great to see that both the data and software are publicly available. I would encourage the authors to also publish their analysis code as that would improve the reproducibility aspects of their work (for example, some software may require parameter inputs that are missing in the main text).

All analysis code has been uploaded to our public github repository (<https://github.com/rosales/VarID2>). We also published the code with a valid DOI on github by using Zenodo: [10.5281/zenodo/7752056](https://zenodo.org/badge/latest/doi/10.5281/zenodo/7752056) (<https://zenodo.org/badge/latest/doi/616373174>).

Minor comments

1. Perhaps you can simplify to use β_j instead of $[\beta']_j$? That would simplify the notation (it is unclear why the ' is needed)

We used the β'_j notation for the cell specific normalization term, to distinguish this from the shape parameter of the Gamma distribution (β_j^t). Since this already has a superscript t, we follow the recommendation of the reviewer and adjust the notation to β_j .

2. The authors use equation (6) to justify why a single parameter is used to model the shape and rate of the Gamma distribution. Whilst this provides a good intuition for this choice, the formal interpretation is a bit more nuisance. In particular, equation (6) refers to the sample average for cell-specific factors that are already inferred using the available data. In that context, the average is indeed exactly equal to 1. However, equation (8) means that the expected value for $[\beta']_j$ is equal to 1.

The expected value for β'_j equals 1 by definition, and since we wanted to model the distribution of β'_j by a Gamma distribution, the mean of this distribution has to be constrained to the expected value of β'_j , i.e., to 1. Since the mean of a Gamma distribution is given by the ratio of shape and rate parameter, equation (8) follows.

3. Page 20 states that a pseudo-count (=1) was added to safeguard against false positive due to outliers. However, as noted by Townes et al (2019), the addition of a pseudo-count may introduce a larger gap between zero and non-zero observations. In such cases, adding a pseudo-count could lead to more extreme outliers. It would be good to discuss how the observation by Townes et al (2019) may affect VarID2 and whether the results are robust to different pseudo-count values.

We thank the reviewer for bringing this point to our attention. To address this concern, we introduced a pseudocount parameter to the pruneKnn function to enable the user to vary this value. As default we chose a value of 1. We then used the bone marrow single-cell RNA-seq data from Tusi et al. [22] to compute noise across all neighborhoods for pseudocount=1 and pseudocount=0.1 and compared noise averaged across all neighborhoods for all genes (Reviewer Fig. 8). This comparison shows a high correlation (Pearson's R=0.99) and only small quantitative differences of noise estimates for the two choices of pseudocounts.

Reviewer Fig. 8: Comparison of noise estimates for all genes averaged across all neighborhoods, computed with pseudocount=1 versus pseudocount=0.1. Bone marrow single-cell RNA-seq data from Tusi et al. [22] were used.

4. Scatter-plots visualising several thousands of dots (such as Fig 1e) can be misleading unless the density of dots is incorporated.

We thank the reviewer for pointing this out, however, we believe that incorporating the density of the dots would not allow to show a direct comparison between different groups. For example, in Fig. 1e of the main manuscript, we want to highlight that noise inflation is reduced with MAP estimation when compared with ML estimation, by overlapping noise estimates in different colors. In Reviewer Fig. 9 (top panel), we show two separate plots with gradient color, corresponding to the density of the dots, however, comparing the corresponding estimates by their location on the x-axis is not so straightforward. In the lower panel of the Reviewer Fig. 9, we include marginal density plots, representing the distribution of the dots along both axes. This can improve the interpretation of the density of dots, but we believe that the added value of this representation is limited and we would prefer to keep the original version of the plots.

Reviewer Fig. 9: ε estimates as a function of the mean on logarithmic scale, obtained from ML or MAP estimation. Top: color gradient indicating dot density. Bottom: different colors represent each inference method, and the distribution of the dots is represented with the marginal density plots.

5. Version numbers are missing for some software packages. The number of iterations is also missing for BASiCS.

We added an additional Table S2 (in Additional_File_1) containing details of the software packages and versions. We also indicated the parameters used for running BASiCS in the methods section as follows:

“We applied BASiCS [18,20] to the simulated dataset by using the implementation with regression model and without spike-in, and choosing suggested parameters for reaching convergence: number of iterations $N=20000$, thinning period length $Thin=20$; and length of burn-in period $Burn=10000$.”

Reviewer #3:

In this paper, the authors aim to distinguish biological variability from technical variability. Their method builds on a previous approach, "VarID", which quantifies variation in gene expression but does not separate out technical effects. Here, the authors present four key results, which are well-supported by experiments and analyses. The results are:

1. For every gene, the authors find that the abundance of nuclear-localized transcripts shows higher biological variation (over-dispersion over Poisson noise) than cytoplasmic transcripts. They suggest that this may be due to nuclear pore transport acting as a buffer.
2. The authors identify two classes of genes with fundamentally distinct noise dynamics: class A genes, in which noise is positively correlated with expression and availability, and class B genes, in which larger expression is associated with less biological variability. They note that class A genes are enriched in genes related to immune activation, which the authors believe is related to the heterogeneity of extrinsic signals that control this process.
3. The authors report that long-term HSCs are lowest in noise (i.e. are more transcriptionally homogeneous) among the hematopoietic progenitors, and aging is associated with higher variability.
4. The authors identify Dlk^+ LT-HSC as a stem cell subset that is identifiable from noise profile, but not from mean transcriptomic differences alone. The authors characterized this population functionally in vivo and in vitro, providing convincing evidence of their role as contributors to age-related myeloid bias.

Overall these are a strong set of results, and we have no major comments. We have some specific technical comments below for additional analyses that will strengthen the claims. We also suggest adding to the discussion of the results in a few places.

We thank the reviewer for the positive comments on our work and appreciate the feedback, which enabled us to improve the manuscript.

1. Methods: Authors should provide a supplementary figure supporting their claim that estimating $\mu_{i,j}$ separately from $\epsilon_{i,j}$ arrives to correlated estimates and is thus unnecessary.

We agree with the reviewer that this is important data to show. We added a panel to Fig. S1 (new Fig. S1g) with a comparison between the results of simultaneous 2D inference of μ and ϵ versus the results from ϵ inference with calculated μ for an exemplary neighbourhood (using murine Kit^+ bone marrow single-cell RNA-seq data from Tusi et al. [22]). The inferred ϵ values of the simultaneous 2D inference correspond almost exactly to the inferred ϵ values from the 1D inference except for a small set of points (see Reviewer Fig. 10 left, highlighted in red). For most of these

points, the inferred mean from the 2D inference is zero, while the calculated mean is significantly larger than zero (see Reviewer Fig. 10 right, highlighted in red), suggesting convergence to a local maximum with mean zero for the 2D fit. We tested different optimization algorithms for maximum likelihood inference (Nelder-Mead, L-BFGS-B) and obtained consistent results.

We added a paragraph to Results explaining these findings:

“We also tested simultaneous MAP inference of both $\mu_{i,j}$ and $\varepsilon_{i,j}$, and found inferred values of $\mu_{i,j}$ to be in good agreement with the calculated average (individual MAP estimation, Fig. S1g). However, for a small percentage of genes simultaneous inference of $\mu_{i,j}$ and $\varepsilon_{i,j}$ results in local minima with vanishing $\mu_{i,j}$ despite a non-zero local average, supporting our robust inference strategy.”

Reviewer Fig. 10: Left: ε estimates inferred by simultaneous 2D inference (μ and ε) versus 1D inference (ε only with calculated μ , individual MAP estimation, following the strategy of VarID2) for an example neighborhood (murine *Kit+* bone marrow single-cell RNA-seq data from Tusi et al. [22]). Outlier points with strongly diverging estimates are highlighted in red. Right: similar to the left plot, but showing μ estimates and the outliers selected from the left panel are highlighted in red. Outliers correspond to data points with vanishing μ estimates, whereas the calculated mean is a positive number, suggesting that 2D inference did not converge to the true value in rare cases. A pseudocount of 10^{-4} and 10^{-2} was added to ε and μ values, respectively.

- Figure 1c/d: How robust are the orderings suggested by the box plots to random gene subsetting? Are the orderings maintained when including only shared highly variable genes? It'd be interesting to see a scatterplot similar to S2f comparing cytoplasmic vs cytoplasmic and nuclear vs nuclear for, e.g., CD4 naive and CD4 memory T cells.

We feel that selecting a subset of genes, i.e., the shared set of highly variable genes would bias our estimates, and would be difficult to interpret, whereas the inclusion of all genes permits a global, unbiased comparison. Since Figure S2 is already quite packed and the information shown by the scatterplots, which the reviewer

recommends is already represented in Fig. S2e, we would not like to add additional scatterplots to this Figure.

3. Figure 2h: This figure would benefit from including the ratio predictions by VarID2, since the sampling noise in the ratio between the nuclear and cytoplasmic noise appears to be very large in both methods, casting doubt on what «excellent agreement» would constitute.

We believe that the figure already shows what the reviewer would like to see. The ratio between nuclear and cytoplasmic noise is shown for the sequencing data, as estimated by VarID2, and for the smFISH-based estimate. To make this clearer, we now added "(VarID2 estimate)" behind "RNA-seq" to the legend.

We further agree with the reviewer, that "excellent agreement" may be an overstatement considering the size of the error bars and adjusted our statement in the Results:

"For these genes, we quantified nuclear and cytoplasmic mRNA counts by smFISH (Fig. 2g and Fig. S2h), and computed the ratio of residual biological noise between nucleus and whole cells, which was consistent with the noise ratios predicted by VarID2 ("Methods" and Fig. 2h)."

4. Figure 3: Figure 3 would benefit from including a subpanel that summarizes the findings of Fig. 3b-c (correlations and specificities) quantitatively. For instance, authors may consider including some visual representation of the distribution of correlations between expression and noise in class A genes vs class B genes.

We thank the reviewer for this suggestion. We show histograms comparing the coefficient of correlations of class A and class B genes in Reviewer Fig. 11. There is a gap in the noise - gene activity correlations between -0.05 and 0.05, since these are the thresholds for determining the different groups of genes.

These histograms were added to Fig. S3, as panel b. Moreover, we adjusted Fig. S3 by removing the original panel b, which contained heat maps of class A gene profiles (expression, gene activity and noise). A representative subset of these class A genes is already shown in Fig. 3b, and thus the heatmaps in the supplementary figure were redundant.

Reviewer Fig. 11: Histograms of the Pearson correlations for class A and class B genes. Left: correlation between gene expression and gene activity. Right: correlation between noise and gene activity.

- Figure 5c: Figure 5c does not show a large effect size in the difference of noise between young B and aged B. Authors should clarify the effect size being small.

We performed a Wilcoxon test to compare noise levels in aged versus young HSCs for each batch to confirm that the differences are significant. We added these results to Fig. 5c and the corresponding description in the figure legend. We also adjusted our statement in the manuscript accordingly:

“For both batches, we observed elevated noise levels in aged versus young LT-HSCs, albeit with limited effect size for batch B (Fig. 5c), indicating that the transcriptome of LT-HSCs becomes more variable with age.”

- In the Discussion: relating to the two distinct modes of gene regulation, can the authors propose additional mechanisms to explain differences in their cell type specificity? And can they (briefly) suggest the nature of follow-up experiments that could test these mechanisms?

We expanded on this topic in the Discussion to propose an explanation for the cell-type specificity of class A genes and potential ways to test this experimentally:

“We speculate that particular cell type-specific functions, such as regulation of immune response, could be evolutionarily selected for to be controlled by class A genes. Such noisy switches could enable a broad spectrum of responses across a population of cells, and feedback mechanisms may lead to selection and clonal expansion of cells with the appropriate response level. In principle, the relevance of noisy responses could be experimentally investigated by replacing switch-like key regulators with more homogeneously-induced alleles, e.g., by changing promoter or enhancer motifs predicted to control expression variability.”

7. In the Discussion: In the analysis of hematopoietic cells, this work raises the intriguing hypothesis that stochasticity in cell cycle gene expression underlies the impaired proliferation of W41/W41 HSCs. In discussion, could the authors propose the type of experiments that might in the future be used to test this hypothesis?

We feel that the experiments conducted in the original publication by Dahlin et al. strongly support this idea, since the same number of colonies were observed when culturing wild-type and W41/W41 HSCs, but an increased number of small colonies for the culture of W41/W41 HSCs was found. This observation suggests, that a fraction of W41/W41 HSCs does not progress to proliferate efficiently. Beyond this it will be very challenging to design experiments proving stochastic cell cycle progression, or stalling of individual cells. Perhaps single-cell life tracking experiments could help to address this point, but since this is not our area of expertise, we do not feel comfortable suggesting these experiments in the Discussion.

Minor

1. Page 5, line 31: “Major sources of this noise component are cell-to-cell differences in sequencing efficiency and cell volume” This statement would benefit from a citation.

We added Grün et al. (2014) Nature Methods as a reference, where this source of noise was explicitly described and modeled.

2. Page 8, line 5: Genes with equal expression in the nuclear and cellular compartment were used for comparison. How was this equality determined?

We added an explanation to the manuscript as follows:

*“We selected candidate genes with **similar** expression in the nuclear and the cellular compartment, **by performing differential expression analysis and keeping only genes without significant change** (Fig. S2f).”*

Moreover, there is a detailed explanation in the legend of Fig. S2f:

*“Differential expression analysis of CD8 naïve T cells (cluster 4 in (d)), comparing **snRNA-seq** versus scRNA-seq samples. Genes were split into ten equally populated bins, based on their mean expression (vertical lines) and genes with no differential expression (red dots) were selected to compare noise levels per gene (see Fig. 2f). Threshold values: $FC > 1.25$, $padj < 0.001$.”*

3. Figure 3: ****Strengthening the lines in the plot showing the N-Pk and Ex-Pk connections would help the patterns be more distinguishable in Figures 3e-f.

We followed the recommendation of the reviewer.

4. Figure 5: Fig. 5 contains cluster labels that are difficult to read. It is suggested that a sub-panel in Fig. S5 be included, showing the boundaries of the clusters and their assigned identities (e.g. LT-HSC, MPP, Other) in a clear manner.

We included cluster labels of the four LT-HSC cell populations.

5. Methods: Page 22, line 8 - Typo: "neighborhood L consists"

We thank the reviewer for spotting this and fixed the typo.

6. Methods: Page 22, line 36 - This line wrongly implies that N_j is a $k+1$ long vector rather than a scalar.

Indeed, N_j is defined as a vector of length $k+1$. We realized that the notation in this section was confusing and made it clearer in the revised manuscript. We thank the reviewer for bringing this to our attention.

7. Methods: Page 22, line 16: "Variability in total UMI counts across nearest neighbor cells are caused by technical cell-to-cell variability in sequencing efficiency and by variations in cell size or RNA content" This statement would benefit from a citation.

We added Grün et al. (2014) Nature Methods as a reference, where this source of noise was explicitly described and modeled.

8. Methods: Reads would benefit from an operational definition of systematic mean-variance dependence as used in this manuscript.

We apologize, but it is unclear to us what kind of definition the reviewer expects. By "systematic mean-variance dependence" we mean any kind of non-random dependence.

9. Methods: Page 26, line 8: What is the explanation for choice of parameters for $\Gamma(2,2)$?

The parameters were chosen such that the global UMI count variability modeled by this distribution corresponds to the effect size observed in real datasets. We expanded this section of the Methods to provide more detailed explanations.

Second round of review

Reviewer 1

The authors have addressed all my comments. In particular, I was happy to see the benchmarking with a different alignment strategy (exons/introns). The concordance of results reflects the method is robust. I believe with the updated vignette (and the reproducible code for this manuscript) the method will see wide adoption. I have no further suggestions.

Reviewer 2

The authors have addressed most of my comments in the revised manuscript. However, a few things remain unresolved.

1. Regarding the batch correction (major comment 2). Thanks for clarifying how batch correction is integrated into VarID2. I agree that noise estimation is best performed within batches, as integration can introduce distortion. However, if a user decides to use Harmony integration, it is still unclear to me how is such data processed. In their response, the authors state that “neighbors are inferred post Harmony integration of the Pearson’s residuals resulting from the GLM-normalization” and that “For both batch integration strategies, pruning of integrated neighborhoods is done based on raw UMI counts in the same way as without batch integration.” but they do not state what data is used as input for their local negative binomial models. Please clarify.

2. Regarding the definition of the Gamma-Poisson mixture (major comment 4). Thanks for editing the description of the model, it helped me to better understand how was VarID2 implemented. A few remaining comments:

a. The jump from eqs (3)+(8) to (9) is not trivial due to the inclusion of the inflation term. However, I can do the transition if I assume that the shape and rate parameters of the Gamma distribution in (8) are equal to $\alpha_j^t / \mathbb{E}[\epsilon^t]_{(i,j)}$, i.e. the variance of β_j is equal to $\mathbb{E}[\epsilon^t]_{(i,j)} / \alpha_j^t$. If this is correct, can the introduction of the inflation term effectively be interpreted as having gene- and cell-specific scaling factors β_j ?

b. The sentence commencing “We further multiply ...” is not required since it follows from the definition in eq (3).

c. When inferring the inflation terms $\mathbb{E}[\epsilon^t]_{ij}$, (eq (13)), is α_j^t fixed by fitting a Gamma model (eq (8)) to empirical scaling factors β_j calculated as per eq (4)? If so, please clarify this on the text. My suggestion would be to (i) introduce the NB formulation, (ii) explain how μ_{ij} and α_j^t are inferred (this will involve explaining how empirical values for β_j are obtained and (iii) explaining the inference for $\mathbb{E}[\epsilon^t]_{ij}$. This will also address minor concern 2 from my previous report, in which I enquired about the difference in interpreting sample means vs expectations for β_j .

Authors Response

Point-by-point responses to the reviewers’ comments:

Reviewer #2:

The authors have addressed most of my comments in the revised manuscript. However, a few things

remain unresolved.

We thank the reviewer for the positive evaluation of our manuscript and for the valuable comments on further required clarifications in our methods section.

1. Regarding the batch correction (major comment 2). Thanks for clarifying how batch correction is integrated into VarID2. I agree that noise estimation is best performed within batches, as integration can introduce distortion. However, if a user decides to use Harmony integration, it is still unclear to me how is such data processed. In their response, the authors state that “neighbors are inferred post Harmony integration of the Pearson’s residuals resulting from the GLM-normalization” and that “For both batch integration strategies, pruning of integrated neighborhoods is done based on raw UMI counts in the same way as without batch integration.” but they do not state what data is used as input for their local negative binomial models. Please clarify.

For the pruning and the noise inference only raw counts are used. We clarified this in the revised manuscript.

“For both batch integration strategies, pruning of integrated neighborhoods and noise inference is done based on raw UMI counts in the same way as without batch integration.”

2. Regarding the definition of the Gamma-Poisson mixture (major comment 4). Thanks for editing the description of the model, it helped me to better understand how was VarID2 implemented. A few remaining comments:

a. The jump from eqs (3)+(8) to (9) is not trivial due to the inclusion of the inflation term. However, I can do the transition if I assume that the shape and rate parameters of the Gamma distribution in (8) are equal to $\alpha_j^t / \varepsilon'_{i,j}$, i.e. the variance of β_j is equal to $\varepsilon'_{i,j} / \alpha_j^t$. If this is correct, can the introduction of the inflation term effectively be interpreted as having gene- and cell-specific scaling factors β_j ?

The Gamma distribution in (8) only accounts for technical variability explained by cell-to-cell variability in total transcript counts. Therefore, we chose to rescale by a factor $1/\varepsilon'_{i,j}$ in

order to account for biological variability. While the scaling factor β_j explaining the technical variability remains only cell specific, the noise inflation term $\varepsilon'_{i,j}$ in general is gene-specific.

We explained this step in the revised manuscript in more detail. See response to point c.

b. The sentence commencing “We further multiply ...” is not required since it follows from the definition in eq (3).

We still felt it was good to mention this step here explicitly and made this sentence clearer in the revised manuscript. See response to point c.

c. When inferring the inflation terms ε'_{ij} , (eq (13)), is α_j^t fixed by fitting a Gamma model (eq (8)) to empirical scaling factors β_j calculated as per eq (4)? If so, please clarify this on the text. My suggestion would be to (i) introduce the NB formulation, (ii) explain how μ_{ij} and α_j^t are inferred (this will involve explaining how empirical values for β_j are obtained and (iii) explaining the inference for ε'_{ij} . This will also address minor concern 2 from my previous report, in which I enquired about the difference in interpreting sample means vs expectations for β_j .

This is correct. In order to clarify this point and to address the concerns raised in a. and b., we rewrote the respective paragraph in the methods section of the revised manuscript:

“We infer the parameter α_j^t by fitting the Gamma distribution in equation (8) to the empirical values of β_j in the local neighborhood of cell j (see equation (4)). To fit a Poisson-Gamma model capturing the technical noise components defined above, and the residual biological variability, we introduce an inflation term $\varepsilon'_{i,j}$ that accounts for the biological variability of gene i in the neighborhood of cell j .

First, in order to obtain a Gamma distribution for the Poisson rate λ_{ij} (see equation (3)), we rescale the shape and the rate parameter α_j^t by $1/\varepsilon'_{ij}$, since α_j^t does not account for biological variability. We further multiply the rescaled rate parameter $\alpha_j^t/\varepsilon'_{ij}$ by $1/\mu_{ij}$ to match the mean of λ_{ij} which equals μ_{ij} by definition (see equation (3)). This procedure yields a Gamma distribution of the Poisson rate λ_{ij} reflecting variability of gene-specific transcript counts:

$$\lambda_{ij} \sim \Gamma\left(\frac{\alpha_j^t}{\varepsilon'_{i,j}}, \frac{\alpha_j^t}{\varepsilon'_{i,j} \cdot \mu_{ij}}\right) \quad (9)$$

Second, the negative binomial distribution for transcript counts $X_{i,j}$ is determined as the corresponding Poisson-Gamma mixture

$$X_{i,j} \sim NB \left(\mu_{i,j}, r_{i,j} = \frac{\alpha_j^t}{\varepsilon'_{i,j}} \right) \quad (10)$$

with $\mu_{i,j}$ indicating the mean transcript counts per gene i across local neighborhoods with central cell j ."